# STACKELBERG COUPLING OF ONLINE REPRESENTATION LEARNING AND REINFORCEMENT LEARNING

**Fernando Martinez**
Fordham University
New York, NY, USA
`fmartinezlopez@fordham.edu`

**Tao Li**[*]
City University of Hong Kong
Hong Kong
`li.tao@cityu.edu.hk`

**Yingdong Lu**
IBM Research
Yorktown Heights, NY, USA
`yingdong@us.ibm.com`

**Juntao Chen**
Fordham University
New York, NY, USA
`jchen504@fordham.edu`

## ABSTRACT

Deep Q-learning jointly learns representations and values within monolithic networks, promising beneficial co-adaptation between features and value estimates. Although this architecture has attained substantial success, the coupling between representation and value learning creates instability as representations must constantly adapt to non-stationary value targets, while value estimates depend on these shifting representations. This is compounded by high variance in bootstrapped targets, which causes bias in value estimation in off-policy methods. We introduce Stackelberg Coupled Representation and Reinforcement Learning (SCORER), a framework for value-based RL that views representation and Q-learning as two strategic agents in a hierarchical game. SCORER models the Q-function as the leader, which commits to its strategy by updating less frequently, while the perception network (encoder) acts as the follower, adapting more frequently to learn representations that minimize Bellman error variance given the leader's committed strategy. Through this division of labor, the Q-function minimizes MSBE while perception minimizes its variance, thereby reducing bias accordingly, with asymmetric updates allowing stable co-adaptation, unlike simultaneous parameter updates in monolithic solutions. Our proposed SCORER framework leads to a bi-level optimization problem whose solution is approximated by a two-timescale algorithm that creates an asymmetric learning dynamic between the two players. Extensive experiments on DQN and its variants demonstrate that gains stem from algorithmic insight rather than model complexity.

## 1 INTRODUCTION

Deep Reinforcement Learning (RL) has achieved outstanding success, exemplified by the superhuman performance of Deep Q-Networks (DQN) in complex environments (Mnih et al., 2015). The dominant approach for value-based methods involves training both the representation and the value function within a single neural network, encouraging a beneficial co-adaptation between the features learned and value estimates, leading to significant breakthroughs (Hessel et al., 2018; Kapturowski et al., 2018); however, this monolithic design is subject to an instability known as the deadly triad (Sutton & Barto, 2018; Van Hasselt et al., 2018). This instability results from the challenging interaction between function approximation using neural networks, bootstrapping, and off-policy learning from data stored in an experience replay buffer, where errors in the current value function are bootstrapped into its own learning targets, leading to a risk of unbounded divergence, an issue first formally demonstrated in Baird (1995). This risk remains a critical practical concern for modern agents; recent work has empirically linked this core instability to phenomena like representation collapse and catastrophic learning failures in deep Q-learning (Lyle et al., 2022).

A prominent line of work seeks to mitigate this instability by directly enhancing the quality of the learned representations. This is often accomplished by extending the main RL objective with auxiliary losses, borrowing from advances in self-supervised and representation learning (Jaderberg et al., 2017; Schwarzer et al., 2021; Laskin et al., 2020b). While this strategy can improve performance,

---

[*]Corresponding author.

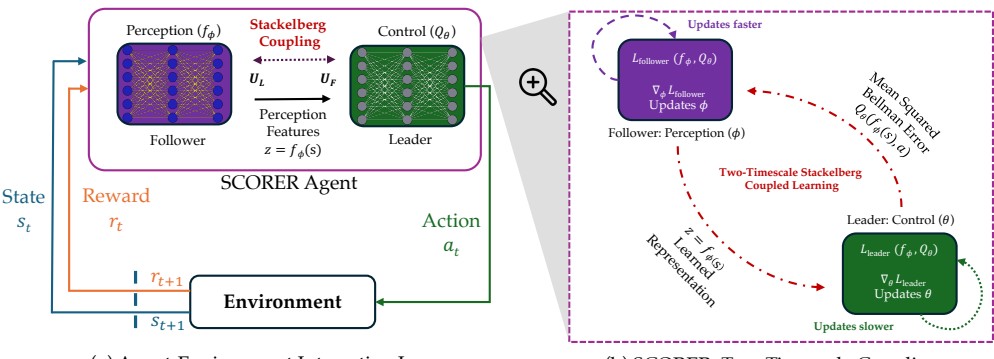

(a) Agent-Environment Interaction Loop  (b) SCORER Two-Timescale Coupling

Figure 1: SCORER framework. **(Left)** Overall agent-environment interaction loop. Internally, the agent comprises a perception network (Follower, $f_\phi$) and a control network (Leader, $Q_\theta$) that interact via Stackelberg game dynamics ($U_F, U_L$ representing their utility functions). The perception network produces features $z = f_\phi(s)$ used by the control network. **(Right)** Details the Stackelberg interaction within the agent.

utilizing a shared network to fulfill multiple objectives can introduce the challenge of conflicting gradients (Yu et al., 2020), wherein the gradient from an auxiliary loss opposes that of the primary value-learning objective. This interference presents a fundamental trade-off, as the effort to stabilize representations introduces a new obstacle to learning the value function itself.

This trade-off suggests that simply augmenting the monolithic optimization is an insufficient solution, motivating a fundamental restructuring of the optimization problem itself. We therefore propose Stackelberg Coupled Representation and Reinforcement Learning (SCORER), a framework that recasts the interaction between perception and control as a hierarchical Stackelberg game (Lambertini, 2018). In this game, the Q-function acts as a slow-updating leader, providing the stable learning target necessary for a fast-adapting perception network (the follower) to learn a robust representations by minimizing Bellman error variance. This objective incentivizes the creation of representations that can handle the noisy targets encountered during exploration, preventing the representation collapse that can afflict monolithic agents (Lyle et al., 2022). This division of labor resolves the underlying trade-off, using temporal separation to create the stability required for this co-optimization.

We create SCORER's game-theoretic dynamics through a practical and computationally efficient algorithm. The Stackelberg equilibrium is approximated using two-timescale gradient descent, where the follower can track a best response to the leader's slowly evolving strategy. Implementing SCORER is simple, as the hierarchical coupling is achieved by assigning the two-players distinct decaying learning rates that satisfy the conditions for two-timescale convergence, with no alterations to the underlying network architectures. This two-timescale dynamic provides a principled mechanism for achieving stable, coordinated adaptation, an approach supported by established theory in stochastic approximation (Borkar, 1997; Fiez et al., 2020).

Figure 1 provides a high-level overview of the SCORER framework. Our code is available at `https://github.com/fernando-ml/SCORER`. In summary, **our contributions are three-fold**: 1) We introduce the *SCORER framework*, a novel game-theoretic formulation that recasts the interaction between perception and control to address the core instabilities of off-policy value-based learning directly. 2) We develop a practical and efficient *two-timescale algorithm* requiring only update frequency modifications, stabilizing the co-adaptation of representations and value functions. 3) We provide *extensive empirical validation*, demonstrating that SCORER consistently improves the sample efficiency and final performance of various off-policy Q-learning agents across multiple benchmarks.

## 2 RELATED WORK

Representation learning has been a pivotal topic in RL, even before the advent of deep neural networks. Early stage efforts concentrated on value function approximation (Thrun & Schwartz, 1993; Tsitsiklis & Roy, 1997) and associated basis function selections (Li & Zhu, 2019; Geramifard et al., 2013). Entering the age of deep learning, RL harnesses the representation power of neural networks and becomes capable of solving high-dimensional complex tasks with multi-modal inputs, such as texts (Li et al., 2016), images (Mnih et al., 2015), and multi-modal sensor data (Liu et al.,

2017; Yin et al., 2024; Li et al., 2025). The confluence of deep representation learning and RL leads to the vibrant research field of deep RL.

Depending on the purpose, representation learning methods in deep RL can be classified into two major categories: those aimed at facilitating training and those designed to enhance testing performance. In the training phase, learning appropriate representations helps with 1) *dimension reduction*: extracting low-dimensional features from high-dimensional inputs, making the RL tasks tractable, a typical example of which is vision-based control tasks (Lesort et al., 2018; Li et al., 2023); 2) *sample efficiency*: capturing underlying environment dynamics and value structure (Subramanian et al., 2022; Lee et al., 2020; Dabney et al., 2021; Vincent et al., 2025), which reduce the number of training episodes and foster faster convergence; and 3) *oriented exploration*: directing future exploration strategies based on past experiences and intrinsic motivation (Kulkarni et al., 2016), particularly in reward-free and sparse-reward settings (Pathak et al., 2017; Hazan et al., 2019). Some other considerations include training *stability* (Greydanus et al., 2018) and *explainability* (Dazeley et al., 2023; Li et al., 2023). Regarding testing improvement, previous work focuses on the transferability of representations for improved generalization to downstream or similar tasks (Agarwal et al., 2021a; Träuble et al., 2021).

Our proposed Stackelberg framework aims to improve *sample efficiency* and *stability* in the training stage. Weighing the three mainstream representation learning methods—supervised, unsupervised, and self-supervised learning—this work opts for the unsupervised approach based on the Mean Squared Bellman Error (MSBE). Supervised learning requires additional labeling to inform the agent of the quality of learned representations (Wang et al., 2024), which is often observed in physical control with partial observability, and the representations need to encode structural information of the environment, such as depth maps from RGB images (Mirowski et al., 2017) and physics principles underlying the wireless sensing inputs (Yin et al., 2024; Li et al., 2025). Distinct from additive value decompositions (Anand & Precup, 2023) or descriptive architectural separation (Garcin et al., 2025), this work explores the benefit of *internal game-theoretic coupling*, creating a hierarchical dependency between representation and control without external learning signals (other than task rewards) to interfere with the strategic interactions between the two learning processes.

A similar argument also explains why we do not consider the incorporation of self-supervised learning, such as contrastive learning (Liu et al., 2021; Stooke et al., 2021; Banino et al., 2022), temporal dynamics and state prediction (Subramanian et al., 2022; Schwarzer et al., 2021), and observation reconstruction (Lange & Riedmiller, 2010; Lesort et al., 2018), and most unsupervised learning, including mutual information (Anand et al., 2019), entropy maximization (Hazan et al., 2019), data augmentation (Yarats et al., 2022; Laskin et al., 2020a), and bisimulation (Zhang et al., 2021; Agarwal et al., 2021a). Our proposed unsupervised representation learning solely relies on MSBE and its sample variance without auxiliary tasks or additional learning signals.

## 3 PRELIMINARY

**Reinforcement Learning and Q-Learning.** We consider an agent interacting with an environment formulated as a Markov Decision Process (MDP), defined by the tuple $(\mathcal{S}, \mathcal{A}, P, R, \gamma)$. Here, $\mathcal{S}$ is the state space, $\mathcal{A}$ is the action space, $P(s'|s, a)$ is the state transition probability, $R(s, a)$ is the reward, and $\gamma \in [0, 1)$ is the discount factor. At each step, the environment transitions from state $s$ to $s'$ and provides a reward $r$; if $s'$ is a terminal state, a *done* signal $d = 1$ is captured, otherwise $d = 0$. The agent's goal is to learn a policy $\pi(a|s)$ that maps states to actions (or distributions over actions) to maximize the expected sum of discounted future rewards from a given state $s_t$: $\mathbb{E}_\pi \left[ \sum_{i=0}^\infty \gamma^i r_{t+i} | s_t \right]$, where $r_{t+i}$ is the reward received $i$ steps after time $t$.

In value-based Reinforcement Learning, the optimal action-value function $Q^*(s, a)$ represents the expected return from taking action $a$ in state $s$ and following the optimal policy thereafter. It satisfies the Bellman optimality equation: $Q^*(s, a) = \mathbb{E}_{s'}[R(s, a) + \gamma \max_{a'} Q^*(s', a')]$. Deep Q-Networks (DQN) approximate $Q^*$ using neural networks trained by minimizing the Mean Squared Bellman Error (MSBE): $\mathcal{L}(\theta) = \mathbb{E}_{(s,a,r,s',d) \sim \mathcal{D}} \left[ (Y - Q_\theta(s, a))^2 \right]$.

To stabilize training, the target value $Y$ is calculated using a separate, periodically updated target network ($Q_{\theta_{\text{target}}}$) that prevents the learning target from fluctuating at every step. For a given transition, the target is defined as $Y = r + \gamma(1 - d) \max_{a'} Q_{\theta_{\text{target}}}(s', a')$.

**Stackelberg Game.** A Stackelberg game is a hierarchical game involving two types of players: a *leader* and one or more *followers* (Osborne, 2004). The leader moves first, choosing an action $a_L$ from its action set $\mathcal{A}_L$. The follower, equipped with the full information on the leader's action $a_L$, chooses its own action $a_F$ from its action set $\mathcal{A}_F$ to maximize its own utility $u_F(a_L, a_F)$, given $a_L$. The basic equilibrium relationship for the Stackelberg game is reflected by the following bi-level optimization problem: $\max_{a_L} U_L(a_L, a_F^*(a_L))$, s.t. $a_F^*(a_L) \in \arg\max_{a_F} U_F(a_L, a_F)$, with $U_L(a_L, a_F)$ being the utility of the leader.

The defining characteristic is the leader's ability to anticipate and influence the follower's decision by committing to a strategy first, steering the game toward a favorable equilibrium. This hierarchical dynamic inspires our proposed SCORER framework.

## 4 STACKELBERG COUPLED REPRESENTATION AND REINFORCEMENT LEARNING

In this section, we formalize the game-theoretic principles outlined previously. We model the value-based interaction between perception and control as a hierarchical Stackelberg game to resolve the trade-off between representation stability and value function learning. This framework provides a clear division of labor, implemented through two distinct network components: a perception network that acts as the game's follower, and a control network that acts as the leader.

### 4.1 THE STACKELBERG GAME FORMULATION

**The Leader: Control Network.** The *Control* network $Q_\theta$ assumes the role of the Stackelberg leader as it defines the primary optimization goal (value estimation). Its objective is to learn an optimal action-value function $Q_\theta(z, a)$, mapping state representations $z$ and actions $a$ to expected cumulative discounted future rewards. As the leader, the control network learns on a slower timescale, providing a stable target for the follower. The leader's objective function, $\mathcal{L}_{\text{leader}}$, is the MSBE. Let $\mathcal{D}$ be the agent's data source. This source can be a replay buffer, as in traditional replay buffer-based methods, or a collection of online trajectories, as in PQN (Gallici et al., 2025). The leader aims to solve:

$$\min_{\theta} \; \mathcal{L}_{\text{leader}}(Q_\theta, f_{\phi^*(\theta)}) \triangleq \mathbb{E}_{(s,a,r,s') \sim \mathcal{B} \subset \mathcal{D}} \Big[ (Y - Q_\theta(f_{\phi^*(\theta)}(s), a))^2 \Big], \tag{1}$$

where $\mathcal{B}$ is a batch of transitions sampled from the data source $\mathcal{D}$, and $Y$ is the corresponding Bellman target value. Here, $\phi^*(\theta)$ represents the parameters the follower would ideally converge to given the leader's choice of $\theta$. The hierarchical structure is established by having the control network commit to its parameters on a slower timescale, while the perception network responds on a faster timescale.

**The Follower: Perception Network.** The *Perception* network $f_\phi$ acts as the Stackelberg follower. Its function is to learn an encoder that maps raw observations $s$ to a latent representation $z = f_\phi(s)$. Given the control network's committed strategy (with $\theta$ treated as fixed via stop-gradient during the follower's update), the follower learns on a faster timescale, allowing it to compute an effective best response to the leader's slowly changing strategy. The follower seeks parameters $\phi$ that minimize its own loss function, $\mathcal{L}_{\text{follower}}$.

SCORER is agnostic to the specific choice of follower objective, allowing for different formulations depending on the desired properties of the learned representations. We investigate multiple objectives for $\mathcal{L}_{\text{follower}}$, including directly minimizing the MSBE and minimizing the variance of Bellman errors. Through extensive empirical evaluation (detailed in Section 5 and Appendix K), we find that minimizing the **variance of Bellman errors** yields superior performance, which echoes the observation that MSBE is biased due to the high variance (Baird, 1995; Wu et al., 2021). For a batch $B$, let $\delta_j(\phi, \theta) = Y_j - Q_\theta(f_\phi(s_j), a_j)$ be the Bellman error for transition $j$, where $Y_j$ is the target value. The follower's objective becomes:

$$\phi^*(\theta) \in \arg\min_{\phi} \; \mathcal{L}_{\text{follower}}(f_\phi, Q_\theta) \triangleq \text{Var}_{j \in B}[\delta_j(\phi, \theta)], \tag{2}$$

where $\text{Var}_{j \in B}[\delta_j] = \frac{1}{|B|} \sum_{j \in B} \delta_j^2 - \left( \frac{1}{|B|} \sum_{j \in B} \delta_j \right)^2$ is the sample variance of the Bellman errors over the batch.

The motivation for this variance-minimization objective is to counteract the pathologies of the deadly triad directly. By focusing on the consistency of the Bellman errors across a batch, the follower learns representations that are more consistent across the batch, making them robust to the noisy targets inherent in TD learning. This promotes stability in the learning process, a trait for mitigating the risk of divergence that was first formally demonstrated in foundational work by Baird (1995). This objective transforms the follower into an active stabilizing agent, as it is incentivized to find representations that make the leader's value predictions more uniform across the data distribution.

## 4.2 APPROXIMATING THE STACKELBERG EQUILIBRIUM VIA TWO-TIMESCALE GRADIENT DESCENT

The Stackelberg game described in Section 4.1 translates to a bi-level optimization problem given by equation 3. The control network aims to optimize its objective $\mathcal{L}_{\text{leader}}$ by choosing its parameters $\theta$, anticipating the follower's optimal response $\phi^*(\theta)$:

$$\min_{\theta} \quad \mathcal{L}_{\text{leader}}(Q_\theta, f_{\phi^*(\theta)}) \quad \text{subject to} \quad \phi^*(\theta) \in \arg\min_{\phi} \mathcal{L}_{\text{follower}}(f_\phi, Q_\theta). \tag{3}$$

Solving such bi-level problems directly in high-dimensional, non-convex settings typical of deep reinforcement learning is challenging, with the primary difficulty arising from computing the gradient of the leader's objective with respect to its parameters $\theta$. The leader's objective $\mathcal{L}_{\text{leader}}(Q_\theta, f_{\phi^*(\theta)})$ depends on $\theta$ both directly (through $Q_\theta$) and indirectly (through the follower's response $\phi^*(\theta)$). Applying the chain rule yields

$$\nabla_\theta \mathcal{L}_{\text{leader}} = \frac{\partial \mathcal{L}_{\text{leader}}}{\partial Q_\theta} \nabla_\theta Q_\theta + \frac{\partial \mathcal{L}_{\text{leader}}}{\partial f_{\phi^*}} \nabla_\theta f_{\phi^*(\theta)}, \tag{4}$$

where the first term is the direct gradient through $Q_\theta$, and the second term $\nabla_\theta f_{\phi^*(\theta)}$ captures the indirect effect through the follower's response.

Computing the indirect gradient term is the central challenge of bi-level optimization. It requires the implicit derivative $\frac{d\phi^*(\theta)}{d\theta}$, which contains second-order information about the follower's optimization landscape. While this term is computationally prohibitive to calculate directly in deep learning contexts (Chen et al., 2023), a body of recent work has developed first-order methods to approximate it (Li et al., 2024; Liu et al., 2022; Li et al., 2022; Hong et al., 2023). However, as our focus is on the game-theoretic coupling, we adopt a more direct solution: we approximate the leader's gradient using the first-order term and converge through a principled timescale separation between the players.

This separation ensures that the faster-learning follower can effectively track the best response to the slower-learning leader. In our framework, this asymmetry is achieved using time-dependent learning rate schedules, $\alpha_{\phi,k}$ and $\alpha_{\theta,k}$, that satisfy the key conditions of two-timescale stochastic approximation (TTSA) theory (Hong et al., 2023). Specifically, both learning rates must decay over training steps $k$, while their ratio must also approach zero, i.e., $\lim_{k \to \infty} \alpha_{\theta,k}/\alpha_{\phi,k} = 0$. This guarantees that the leader learns on a sufficiently slower timescale than the follower, allowing the system to converge to a first-order stationary point of the game, as detailed in Algorithm 1. The practical update rules for the follower and leader are thus given by:

$$\phi_{k+1} \leftarrow \phi_k - \alpha_{\phi,k} \nabla_\phi \mathcal{L}_{\text{follower}}(\phi_k; B_{\text{follower}}, Y, \overline{\theta_k}), \tag{5}$$

$$\theta_{k+1} \leftarrow \theta_k - \alpha_{\theta,k} \nabla_\theta \mathcal{L}_{\text{leader}}(\theta_k; B_{\text{leader}}, Y, \overline{\phi_{k+1}}), \tag{6}$$

where $\bar{\cdot}$ (e.g., $\overline{\theta_k}$) denotes a stop-gradient operation that treats the variable as a constant with respect to the gradient calculation, guaranteeing that no gradients flow from one player's update to the other's parameters. The follower first updates to $\phi_{k+1}$ based on the fixed leader parameters $\theta_k$. The leader then immediately updates to $\theta_{k+1}$ using the follower's new state $\phi_{k+1}$ as a fixed input.

Our approach of having the leader optimize based on the follower's recent state, rather than its fully converged response, is a common and practical technique for approximating bi-level optimization (Fiez et al., 2020; Petrulionytė et al., 2024). Under standard smoothness assumptions

for two-timescale stochastic approximation, including Lipschitz continuous gradients and a weaker convexity condition known as the Restricted Secant Inequality (RSI) (Karimi et al., 2016), our method converges to a first-order stationary point of the game Li et al. (2024). This convergence is general and holds for both the MSBE and Bellman Error Variance objectives, as both measure temporal difference consistency.

We provide a detailed convergence analysis in Appendix M, which explicitly characterizes the convergence rates in terms of the size of neural networks, as well as the learning rates schedules.

### 4.3 TWO-TIMESCALE STACKELBERG COUPLED LEARNING (SCORER)

The implementation details of SCORER for replay-based agents are detailed in Algorithm 1. Both the perception network (follower) and the control network (leader) are updated at the same interval, $T_{\text{Update}}$, where the timescale separation that approximates the Stackelberg dynamic is achieved by setting the two players distinct, time-dependent learning rate schedules, $\alpha_\phi$ and $\alpha_\theta$. In line with two-timescale convergence theory, the schedules decay over the course of training and are chosen such that the follower's learning rate, $\alpha_{\phi,k}$, remains significantly larger than the leader's, $\alpha_{\theta,k}$, making sure that their ratio diminishes over time. We model the two-player game by stopping the gradient flow during backpropagation within a single update step, so the follower first updates its parameters to $\phi_{k+1}$ based on the fixed leader parameters $\theta_k$, and subsequently, the leader updates its parameters to $\theta_{k+1}$ using the follower's newly updated state $\phi_{k+1}$, completing one step of the hierarchical game.

---

**Algorithm 1** SCORER (Stackelberg Coupled Representation Learning) for Deep Q-Learning Variants

**Require:** Perception $f_\phi$, Control $Q_\theta$; Parameters $\phi, \theta$; Target networks $\phi_{\text{target}}, \theta_{\text{target}}$
**Require:** Learning rates $\alpha_\phi, \alpha_\theta$; Replay buffer $\mathcal{D}$; Minibatch size $N_{\text{batch}}$; Exploration strategy $\mathcal{E}$
**Require:** Total timesteps $T$; Update interval and start step $T_{\text{Update}}, T_{\text{target}}, T_{\text{start}}$; Polyak rate $\tau$
 1: **Initialize** online params $\phi, \theta$; target params $\phi_{\text{target}} \leftarrow \phi, \theta_{\text{target}} \leftarrow \theta$; replay $\mathcal{D}$
 2: Reset env. and get initial state $s$; $t_{\text{env}} \leftarrow 0$
 3: **while** $t_{\text{env}} < T$ **do**                                              ▷ Main environment interaction loop
 4:     $z \leftarrow f_\phi(s)$                                                          ▷ Extract representation
 5:     Select action $a \sim \mathcal{E}(Q_\theta(z, \cdot))$
 6:     Execute $a$, observe $r, s', d$; Store $(s, a, r, s', d)$ in $\mathcal{D}$
 7:     $s \leftarrow s'$; $t_{\text{env}} \leftarrow t_{\text{env}} + 1$
 8:     **if** $t_{\text{env}} > T_{\text{start}}$ **then**
 9:         **if** $t_{\text{env}} \pmod{T_{\text{Update}}} = 0$ **then**
10:             Sample minibatch $B_{\text{Follower}}$ from $\mathcal{D}$                ▷ Perception (Follower) Update
11:             Compute targets $Y$ for $B_{\text{follower}}$ using $f_{\phi_{\text{target}}}, Q_{\theta_{\text{target}}}$ (and $Q_\theta$ for DDQN-like variants)
12:             $\phi \leftarrow \phi - \alpha_\phi \nabla_\phi \mathcal{L}_{\text{follower}}(\phi; B_{\text{follower}}, Y, \overline{\theta})$     ▷ Best response to fixed leader
13:             Sample minibatch $B_{\text{leader}}$ from $\mathcal{D}$                      ▷ Control (Leader) Update
14:             Compute targets $Y$ for $B_{\text{leader}}$ using $f_{\phi_{\text{target}}}, Q_{\theta_{\text{target}}}$ (and $Q_\theta$ for DDQN-like variants)
15:             $\theta \leftarrow \theta - \alpha_\theta \nabla_\theta \mathcal{L}_{\text{leader}}(\theta; B_{\text{leader}}, Y, \overline{\phi})$
16:         **end if**
17:         **if** $t_{\text{env}} \pmod{T_{\text{target}}} = 0$ **then**                       ▷ Target Network Updates
18:             $\phi_{\text{target}} \leftarrow \tau\phi + (1 - \tau)\phi_{\text{target}}$;    $\theta_{\text{target}} \leftarrow \tau\theta + (1 - \tau)\theta_{\text{target}}$
19:         **end if**
20:     **end if**
21: **end while**
    ***Note:*** *For recurrent variants (e.g., R2D2), $f_\phi$ maintains hidden states $h$ across timesteps, resetting when episodes terminate. The replay buffer stores sequences rather than individual transitions.*
    ***Note:*** *For agents like PQN that do not use a replay buffer, the asymmetric dynamic is achieved using a higher learning rate for the follower without sampling from a replay buffer.*

---

**SCORER's Generality**    While Algorithm 1 details the implementation for replay-based agents, the SCORER principle of a two-timescale, leader-follower dynamic is more general. As shown in our Section 5, it can be seamlessly adapted to modern agents like PQN that do not use a replay buffer.

## 5 EXPERIMENTS AND RESULTS

In this section, we empirically evaluate the performance of SCORER when integrated with multiple established value-based RL algorithms: DQN and its variants (Double, Dueling) (Mnih et al., 2015;

Van Hasselt et al., 2016; Wang et al., 2016), and R2D2 (Kapturowski et al., 2018) for partially observable environments. We also compare it against the Parallelized Q-Network (PQN) (Gallici et al., 2025), a recent, high-performance off-policy value-based method that forgoes the use of a replay buffer. Our evaluation covers diverse domains, including the MinAtar suite to measure sample efficiency, the Atari-5 benchmark for high-dimensional visual control tasks, and MinGrid environments that test partially observable scenarios. Additionally, we investigate SCORER's stability properties on challenging counterexamples known to cause divergence in standard TD learning, as well as its robustness to environmental stochasticity. Our primary goal is to assess whether the proposed Stackelberg coupling (Algorithm 1) improves learning compared to standard end-to-end training of these baseline methods without increasing computational overhead or architectural size.

All experiments use an efficient JAX-based framework inspired by PureJaxRL and CleanRL algorithms (Lu et al., 2022; Huang et al., 2022). To provide a fair comparison, the underlying network architectures and hyperparameter budgets are kept identical between SCORER variants and their respective baselines. Consequently, any observed performance gains are directly attributable to the dynamics introduced by SCORER. A detailed description of experimental setups and hyperparameters is in Appendix C.

## 5.1 PERFORMANCE ON GYMNAX ENVIRONMENTS

For the first evaluation, we use SCORER across the MinAtar suite (Young & Tian, 2019) through Gymnax (Lange, 2022). Learning curves comparing the SCORER-enhanced variants against their respective baselines are displayed in Figure 2 over Asterix, Breakout, Freeway, and SpaceInvaders. Table 1 complements these curves by summarizing final performance.

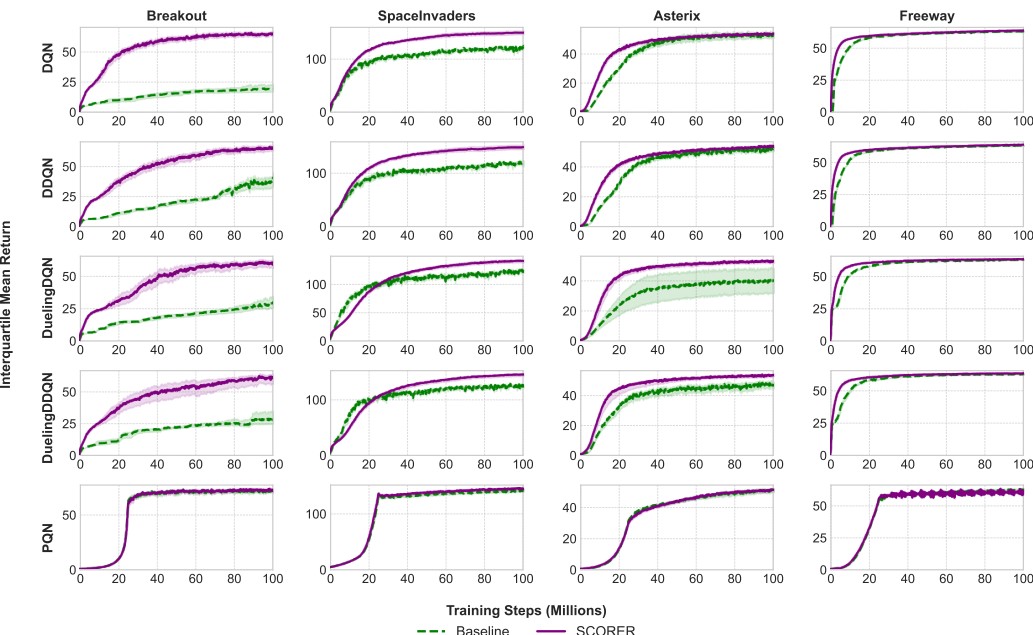

Figure 2: Learning curves on MinAtar comparing SCORER variants against baselines. SCORER generally demonstrate improved sample efficiency and performance across several algorithm-environment combinations.

The results demonstrate that SCORER provides a consistent and significant performance benefit across all tested base algorithms. For the classic DQN family, the SCORER variants show dramatic gains in sample efficiency and final performance. In Breakout, for example, SCORER triples the final score of a standard DQN agent. More importantly, SCORER allows these replay-based methods to become competitive with, and in some cases (e.g., SpaceInvaders), superior to the state-of-the-art methods like PQN baseline.

The final rows of Table 1 show that SCORER's benefits are not limited to replay-based agents. When applied to PQN, our framework provides further statistically significant performance gains. This

Table 1: Final IQM return ($\pm$ 95% Bootstrap CI over 30 seeds) on MinAtar, averaged over the last 10% of training. Highlighted values indicate the best-performing variant within each algorithm family per environment. Highlighted performances are not statistically worse than the best (bootstrap difference test, $p \geq 0.05$).

| | Variant | Asterix | Breakout | Freeway | SpaceInvaders | Speed Comparison |
|---|---|---|---|---|---|---|
| DQN | Baseline | 54.95 ± 0.92 | 19.16 ± 1.50 | 62.70 ± 0.17 | 127.78 ± 0.55 | 1.00x |
| | SCORER | 54.78 ± 0.42 | 65.69 ± 2.52 | 63.03 ± 0.08 | 148.71 ± 2.02 | 0.99x |
| DQN | Baseline | 52.55 ± 2.27 | 19.21 ± 3.81 | 62.23 ± 0.13 | 118.41 ± 2.08 | 1.00x |
| | SCORER | 52.88 ± 0.41 | 63.93 ± 1.86 | 62.75 ± 0.09 | 147.82 ± 4.44 | 0.99x |
| DDQN | Baseline | 50.77 ± 1.19 | 36.47 ± 5.04 | 62.22 ± 0.07 | 116.72 ± 3.36 | 1.00x |
| | SCORER | 52.59 ± 0.27 | 64.44 ± 2.48 | 62.68 ± 0.08 | 146.67 ± 5.14 | 1.00x |
| DuelingDQN | Baseline | 39.22 ± 9.24 | 27.81 ± 3.99 | 61.89 ± 0.09 | 121.21 ± 1.70 | 1.00x |
| | SCORER | 52.28 ± 0.55 | 60.04 ± 2.62 | 62.27 ± 0.09 | 139.08 ± 3.42 | 1.01x |
| DuelingDDQN | Baseline | 46.53 ± 3.05 | 27.68 ± 5.55 | 61.58 ± 0.14 | 122.26 ± 1.44 | 1.00x |
| | SCORER | 52.44 ± 0.24 | 59.92 ± 3.83 | 62.22 ± 0.08 | 142.71 ± 2.82 | 1.00x |
| PQN | Baseline | 50.41 ± 1.34 | 69.97 ± 0.68 | 61.77 ± 0.34 | 137.73 ± 1.04 | 1.00x |
| | SCORER | 50.80 ± 0.62 | 71.16 ± 0.84 | 61.05 ± 0.47 | 141.85 ± 0.76 | 1.00x |

demonstrates that SCORER is not only restricted to replay buffer-based agents but represents an advancement in the co-adaptation of representation and control in value-based RL. Additional results on classic control benchmarks in Appendix H further corroborate this trend.

## 5.2 PERFORMANCE ON ATARI

To efficiently evaluate SCORER's scalability to high-dimensional visual environments, we evaluate on Atari-5 (Bellemare et al., 2013; Aitchison et al., 2023), a statistically validated subset of the Arcade Learning Environment (ALE) that provides a reliable estimate of performance across the full Atari-57 suite while requiring significantly fewer computational resources. We integrate SCORER with PQN (Gallici et al., 2025), being an SOTA algorithm that serves as a strong baseline.

Following standard protocols Machado et al. (2018), we train on each game for 200M frames using 5 independent random seeds per game. We compare vanilla PQN against SCORER PQN, using identical hyperparameters and network architectures for both variants to guarantee that any performance differences are attributable exclusively to SCORER's game-theoretic coupling mechanism. Results in Figure 3 indicate that in BattleZone, DoubleDunk, and NameThisGame, SCORER PQN statistically matches baseline performance, while in Phoenix and Qbert outperforms the backbone model without adding computational overhead. This confirms that our game-theoretic coupling scales effectively to high-dimensional visual control without requiring additional architectural complexity.

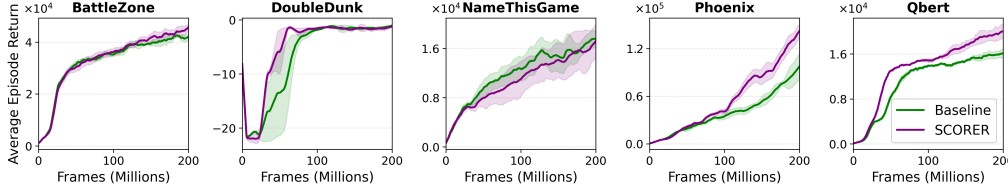

Figure 3: Atari-5 Average Episodic Return

## 5.3 PERFORMANCE ON MINIGRID ENVIRONMENTS

We test SCORER on Min-Grid (Chevalier-Boisvert et al., 2023), a suite of gridworld environments testing navigation, object interaction, and dynamic obstacle avoidance, accessed through Navix (Pignatelli et al., 2024). As these environments are partially observable, we compare R2D2 and its SCORER R2D2 counterpart. Table 4 shows time-to-threshold analysis measuring timesteps to reach

| Environment | Max. Perf | SCORER TTT | SCORER SR | R2D2 TTT | R2D2 SR |
|---|---|---|---|---|---|
| DoorKey 6x6 | 1.00 | 0.3 ± 0.0 | 100% | 0.4 ± 0.0 | 100% |
| SimpleCrossing S9N2 | 1.00 | 0.3 ± 0.0 | 100% | 0.4 ± 0.0 | 100% |
| DistShift2 | 1.00 | 0.1 ± 0.0 | 100% | 0.1 ± 0.0 | 100% |
| LavaGap S6 | 1.00 | 0.1 ± 0.0 | 100% | 0.2 ± 0.0 | 100% |
| GoToDoor 8x8 | 1.00 | 0.2 ± 0.0 | 100% | 0.3 ± 0.0 | 100% |
| Empty Random 8x8 | 1.00 | 0.1 ± 0.0 | 100% | 0.3 ± 0.1 | 100% |
| Dynamic Obstacles 6x6 | 1.00 | 0.5 ± 0.0 | 83% | 0.7 ± 0.1 | 73% |
| Four Rooms | 1.00 | 0.6 ± 0.0 | 97% | - | 0% |

Figure 4: Time-to-threshold analysis showing mean timesteps (in millions) $\pm$ 95% confidence interval to reach 99% maximum performance over 30 seeds. SR (%) is the success rate of runs reaching the threshold.

99% maximum performance within one million timesteps. **SCORER consistently outperforms R2D2, achieving 25-50% faster convergence across all environments. The benefit of the Stackelberg coupling is particularly pronounced in tasks requiring adaptation and exploration.** In Dynamic Obstacles, SCORER improves the success rate by 10 points over the baseline, and in the Four Rooms environment, SCORER achieves a 97% success rate while the baseline R2D2 completely fails to solve the task. Full learning curves in Appendix I.

## 5.4 STABILITY ANALYSIS: BAIRD'S COUNTEREXAMPLE AND STOCHASTIC ENVIRONMENTS

To better understand SCORER's stability properties, we test it against Baird's counterexample (Baird, 1995), a challenging domain known to induce divergence under off-policy learning with function approximation, and against Stochastic Deep Sea (Osband et al., 2020), which combines complex exploration with intrinsic environment noise.

**Baird's Counterexample:** In this setting, we instantiate SCORER with linear networks that match the representational capacity of the standard linear TD baseline, and we retain the original off-policy behavior policy. Figure 5 (**Left**) reports the loss on a logarithmic scale. As expected, the baseline Linear TD method diverges. Conversely, SCORER's variants (using either the MSBE or BE Variance follower objective) stay stable and converge, remarking the effect of the Stackelberg structure. Here, the perception learns representations that are explicitly shaped by the leader's learning dynamics, thus disrupting the destructive feedback loop that causes divergence in the monolithic agent.

**Deep Sea:** Figures 5 (**Center & Right**) evaluate SCORER on both deterministic and stochastic Deep Sea (depth 10) using Bootstrapped DQN (Osband et al., 2016). For each configuration, we run 30 seeds and report the fraction that reach a solved state (Osband et al., 2020). In the deterministic case, both SCORER variants achieve near-complete solved rates before the baseline, indicating faster and more reliable learning. This advantage carries over to the stochastic version as SCORER is able to reach near-complete solved rates, while the baseline plateaus around 20%.

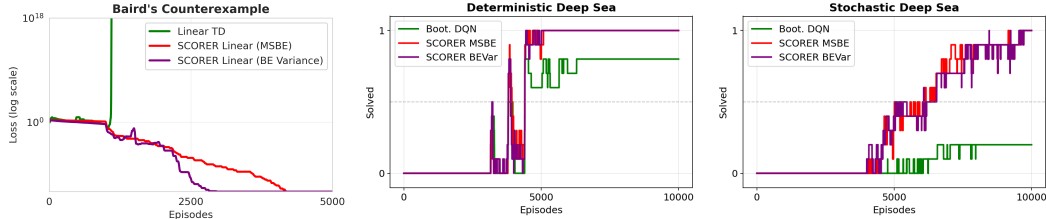

Figure 5: SCORER Stability and Stochasticity Robustness. (**Left**) Baird's Counterexample: The standard SCORER hierarchy outperforms the inverted role configuration. (**Center & Right**) Deterministic & Stochastic Deep Sea: SCORER variants solve the task while baseline solves it ~20%.

## 5.5 DISSECTING THE SCORER FRAMEWORK

To validate the core design of SCORER, we perform a series of ablation studies investigating on the MinAtar suite. We investigate our three central design choices: the follower's objective, the assignment of Stackelberg roles, and the hierarchical coupling dynamic. The main results are presented here as aggregated Interquartile Mean (IQM) scores, while detailed per-environment learning curves for these studies can be found in Appendix K.

**Stackelberg roles:** We test our role assignment by inverting the hierarchy, making perception the leader and control the follower. Figure 6 (**left**) shows that this inversion leads to a collapse in performance, falling well below the monolithic baseline. This result strongly supports our claim that the control objective must lead the representation learning process.

**Follower's objective:** Figure 6 (**center**) shows that using a Bellman Error Variance objective for the follower substantially outperforms a variant where the follower minimizes MSBE. This confirms that

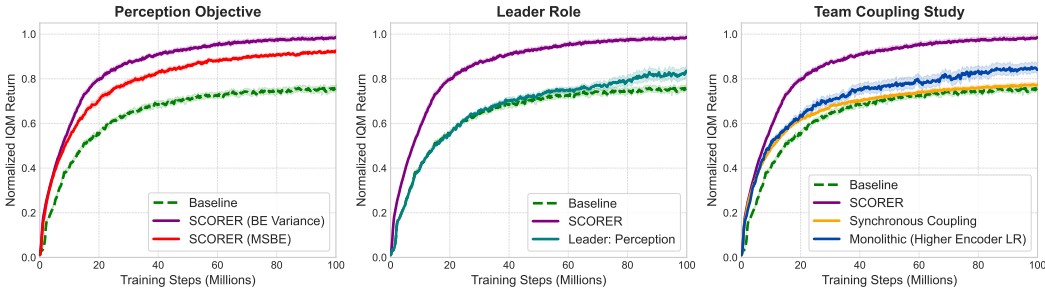

Figure 6: Ablation studies on SCORER's core components on MinAtar. **(Left)** Stackelberg Roles: The standard SCORER hierarchy outperforms the inverted role configuration. **(Center)** Follower's Objective: Bellman Error (BE) Variance is superior to MSBE. **(Right)** Coupling Dynamic: SCORER's hierarchical coupling is critical for performance, outperforming baselines.

an explicit stabilization objective is critical for SCORER's performance, providing a clear advantage over a redundant performance objective.

**Hierarchical vs. Synchronous Coupling:** Finally, we study whether SCORER's benefits stem from its full hierarchical dynamic or merely from the architectural separation of networks. For this, we design a baseline called Synchronous Coupling, where distinct perception ($f_\phi$) and control ($Q_\theta$) networks are updated at the same frequency, with both minimizing MSBE. Stop-gradients operations ensure the updates are independent ($\nabla_\phi \mathcal{L}_{\text{MSBE}}(f_\phi, \overline{Q_\theta})$ and $\nabla_\theta \mathcal{L}_{\text{MSBE}}(\overline{f_\phi}, Q_\theta)$), removing any strategic interaction. This setup cleanly isolates the effect of using two networks from the timescale-based hierarchy of the Stackelberg game. We also test a simpler baseline, a standard monolithic DQN with per-layer learning rates matching SCORER's timescale ratio, where the encoder layers update faster than Q-head layers without architectural changes or stop-gradients.

The results present a clear picture of SCORER's effect (Figure 6, **right**). We observe that the Synchronous Coupling baseline provides marginal benefit, though not significant, and the monolithic per-layer LR baseline does indeed outperform standard DQN, validating that faster updates for the representation are beneficial. Yet, SCORER outperforms both alternatives, and the performance gap is most notable in Breakout and SpaceInvaders (see Appendix K for per-environment results). Overall, the ablations provide evidence that all three design choices (the hierarchical structure, the specific form of the stabilization objective, and the correct assignment of the roles) are important and complementary ingredients of the SCORER framework.

## 6 CONCLUSION

This work revisited and addressed the foundational instability that arises from the tight coupling of representation and control in monolithic value deep Q-learning agents. We present the Stackelberg Coupled Representation and Reinforcement Learning (SCORER), a framework that reframes this dynamic as a hierarchical Stackelberg game. We model the control network ($Q_\theta$) as the leader and the perception network ($f_\phi$) as the follower. This game-theoretic interaction is achieved through a practical two-timescale algorithm where the leader's slower learning rate delivers a stable target, enabling the follower to learn good representations by minimizing Bellman error variance. SCORER consistently exhibited improvements in sample efficiency and/or stabilized returns of a wide range of replay-based and online Q-learning agents, validating the efficacy of its design.

Despite our promising results, some limitations provide clear directions for future work. The current work focused on value-based methods in discrete action spaces, so extending the SCORER principle to actor-critic algorithms for continuous control is an immediate next step. Additionally, our theoretical analysis, which provides convergence guarantees for the two-timescale approximation, could be extended to explore the algorithm's sample complexity, specifically to mathematically characterize how the follower's stabilization objective reduces the number of samples required to learn a high-quality policy. Finally, investigating alternative follower objectives derived from information-theoretic principles presents another compelling research direction. We believe SCORER provides a flexible foundation for developing a new class of more stable and efficient deep RL systems.

ACKNOWLEDGMENTS

Juntao Chen acknowledges support from the Fordham AI Research (FAIR) Grant, the Fordham–IBM Research Award, and the National Science Foundation under Grant ECCS-2138956.

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

APPENDIX

## A   CODE AVAILABILITY

The complete source code for SCORER and the experiments presented in this paper is available at `https://github.com/fernando-ml/SCORER/` to ensure reproducibility. Our implementation is built in JAX (Bradbury et al., 2018) and leverages significant components, particularly for environment vectorization and training loops, from the PureJaxRL framework (Lu et al., 2022)[1]. While core architectural choices (Section C) and hyperparameters for baseline algorithms are consistent with PureJaxRL defaults for MinAtar and DQN, our SCORER-specific mechanisms (perception network objective, perception learning rate) are additions.

## B   LLM USAGE

Writing and editing assistants, including a large language model (LLM) and automated grammar-checking tools, were used to improve the clarity, conciseness, and grammatical correctness of this work. The usage of these tools was strictly limited to polishing the written text. All scientific contributions, including the framework ideation, conceptualization, theoretical analysis, and experimental results, are the original work of the authors.

## C   EXPERIMENT SETUP AND HYPERPARAMETERS

**Software** We used the following software versions:

- Python 3.10 - Python Software License `https://docs.python.org/3/license.html`
- CUDA 12.4 - NVIDIA Software License Agreement `https://docs.nvidia.com/cuda/eula/index.html`
- Jax 0.4.28 - Apache License 2.0 `https://github.com/jax-ml/jax`
- Flashbax 0.1.3 - Apache License 2.0 `https://github.com/instadeepai/flashbax`
- Chex 0.1.90 - Apache License 2.0 `https://github.com/google-deepmind/chex`
- Optax 0.2.5 - Apache License 2.0 `https://github.com/google-deepmind/optax`
- flax 0.10.4 - Apache License 2.0 `https://github.com/google/flax`
- Gymnax 0.0.9 - Apache License 2.0 `https://github.com/RobertTLange/gymnax`
- Navix 0.7.4 - Apache License 2.0 `https://epignatelli.com/navix/`
- Rlax 0.1.7 - Apache License 2.0 `https://github.com/google-deepmind/rlax`
- Envpool 0.8.4 - Apache License 2.0 `https://github.com/sail-sg/envpool`
- OpenAI Gym - 0.26.2 - MIT License `https://github.com/openai/gym`
- PureJaxRL - Apache License 2.0 `https://github.com/luchris429/purejaxrl`

All experiments were conducted on NVIDIA Tesla V100-PCIE-32GB GPUs. A typical experimental run, consisting of training one algorithm variant (e.g., SCORER DQN) over 30 random seeds for $10^8$ total environment time steps on a MinAtar environment, completed in approximately 20 to 27 minutes.

---

[1] `https://github.com/luchris429/purejaxrl/tree/main`

**Hyperparameter Tuning Methodology**    For a fair and rigorous comparison, we performed a two-stage hyperparameter search where initially, for each base algorithm (e.g., DQN, PQN), we performed a systematic grid search to identify its strongest possible configuration on our benchmarks. Then, we inherited these optimal hyperparameters for each SCORER variant and conducted a targeted search for the introduced SCORER hyperparameter (the follower's learning rate, $\alpha_\phi$), while satisfying the timescale separation condition ($\alpha_\phi > \alpha_\theta$).

**SCORER-Specific Hyperparameters**    The key parameters introduced by SCORER are the learning rates for the leader ($\alpha_\theta$) and the follower ($\alpha_\phi$). As described in Section 4, the leader's learning rate ($\alpha_\theta$) is set to the optimal learning rate found for the baselines. The follower's learning rate ($\alpha_\phi$) is then tuned from a set of values greater than $\alpha_\theta$ to establish the necessary timescale separation.

**Architectural Parity**    To confirm that performance gains are a result of SCORER and not increased model capacity, we keep strict architectural parity between each baseline and its corresponding SCORER version. For a given monolithic baseline (e.g., a DQN with a three-layer MLP), the SCORER version is constructed by splitting this same architecture. The initial layers form the perception network ($f_\phi$), and the final layer forms the control network ($Q_\theta$). This way, we ensure that the total number of layers, hidden units, and learnable parameters is nearly identical between the baseline and SCORER agent, isolating the algorithmic contribution.

**Learning Rate Schedule**    For all experiments, learning rates follow a linear decay schedule from their initial values to zero over the full training duration, i.e., $\alpha(t) = \alpha_0(1 - t/T)$ where $T$ is the total number of updates. This schedule is applied identically to both baseline agents and SCORER variants. The Leader network uses the exact same learning rate trajectory as the corresponding baseline, and the Follower uses the same decay schedule starting from a higher value to satisfy the timescale condition.

**Gradient Clipping**    Gradient clipping via global norm is applied identically across all experiments. For baseline agents, gradients are clipped to a maximum norm of 0.5 (MinAtar DQN variants, MinAtar PQN, R2D2) or 0.3 (classic control), and 5.0 for Full Atari. For SCORER variants, the same clipping threshold is applied independently to both the Leader and Follower networks.

Table 2: General Training Hyperparameters for Q-Learning Methods - MinAtar

| Parameter | Value |
|---|---|
| *Training Configuration* | |
| Number of parallel environments | 128 |
| Total timesteps | $1 \times 10^8$ |
| Learning starts (time steps) | $1 \times 10^4$ |
| Training interval (env steps) | 4 |
| *Replay Buffer* | |
| Buffer size | $1 \times 10^5$ |
| Batch size | 64 |
| *Exploration (Epsilon-Greedy)* | |
| $\epsilon$ start | 1.0 |
| $\epsilon$ finish | 0.01 |
| $\epsilon$ anneal time (env steps) | $2.5 \times 10^5$ |
| *Learning Parameters* | |
| Optimizer | Adam |
| Discount factor ($\gamma$) | 0.99 |
| Linear learning rate decay (Baseline & SCORER) | True |
| Target network update interval (env steps) | $1 \times 10^3$ |
| Soft update parameter ($\tau$) for target nets | 1.0 |
| Q-network learning rate ($\alpha_\theta$) | $1 \times 10^{-4}$ |
| Q-networks max gradient norm | 0.5 |

Table 3: SCORER-Specific Hyperparameters - MinAtar

| Parameter | Value |
|---|---|
| *Leader-Follower Architecture* | |
| Optimizer | Adam |
| Leader (control) learning rate ($\alpha_\theta$) | $1 \times 10^{-4}$ |
| Follower (perception) learning rate ($\alpha_\phi$) | $5 \times 10^{-4}$ |
| Max gradient norm for Leader & Follower | 0.5 |

Table 4: General Training Hyperparameters for Q-Learning Methods for classic control environments

| Parameter | Value |
|---|---|
| *Training Configuration* | |
| Number of parallel environments | 10 |
| Total timesteps | $1 \times 10^6$ |
| Learning starts (time steps) | $1 \times 10^3$ |
| Training interval (env steps) | 10 |
| *Replay Buffer* | |
| Buffer size | $5 \times 10^4$ |
| Batch size | 64 |
| *Exploration (Epsilon-Greedy)* | |
| $\epsilon$ start | 1.0 |
| $\epsilon$ finish | 0.01 |
| $\epsilon$ anneal time (env steps) | $2.5 \times 10^5$ |
| *Learning Parameters* | |
| Optimizer | Adam |
| Discount factor ($\gamma$) | 0.99 |
| Linear learning rate decay (Baseline & SCORER) | True |
| Target network update interval (env steps) | $1 \times 10^3$ |
| Soft update parameter ($\tau$) for target nets | 1.0 |
| Q-network learning rate ($\alpha_\theta$) | $1 \times 10^{-4}$ |
| Q-networks max gradient norm | 0.3 |

Table 5: SCORER-Specific Hyperparameters (Classic Control).

| Parameter | Value |
|---|---|
| *Leader-Follower Architecture* | |
| Optimizer | Adam |
| Leader (control) learning rate ($\alpha_\theta$) | $1 \times 10^{-4}$ |
| Follower (perception) learning rate ($\alpha_\phi$) | $3 \times 10^{-4}$ |
| Max gradient norm for Leader & Follower | 0.3 |

Table 6: General Training Hyperparameters for SCORER PQN (MinAtar environment defaults).

| Parameter | Value |
|---|---|
| *Training Configuration* | |
| Number of parallel environments | 128 |
| Total timesteps | $1 \times 10^8$ |
| Rollout length (experience collection) | 32 |
| Training epochs per rollout | 2 |
| Minibatch count per epoch | 32 |
| *Exploration (Epsilon-Greedy)* | |
| $\epsilon$ start | 1.0 |
| $\epsilon$ finish | 0.01 |
| $\epsilon$ anneal period | 40% of training updates ($4 \times 10^7$ steps) |
| *Learning Parameters* | |
| Optimizer | RAdam |
| Discount factor ($\gamma$) | 0.99 |
| Lambda for TD($\lambda$) returns ($\lambda$) | 0.65 |
| Linear learning rate decay (Baseline & SCORER) | True (over first 50% of training) |
| Target network | N/A (uses $\lambda$-returns) |
| Q-network learning rate ($\alpha_\theta$) | $2.5 \times 10^{-4}$ |
| Q-network max gradient norm | 5.0 |

Table 7: SCORER-Specific Hyperparameters for PQN (MinAtar environment defaults).

| Parameter | Value |
|---|---|
| *Leader-Follower Architecture* | |
| Optimizer | RAdam |
| Leader (Q-Network) learning rate ($\alpha_\theta$) | $2.5 \times 10^{-4}$ |
| Follower (Perception) learning rate ($\alpha_\phi$) | $5 \times 10^{-4}$ |
| Max gradient norm for Leader & Follower | 5.0 |

Table 8: General Training Hyperparameters for SCORER PQN (Atari environment defaults).

| Parameter | Value |
|---|---|
| *Training Configuration* | |
| Number of parallel environments | 128 |
| Total timesteps | $5 \times 10^7$ |
| Rollout length (experience collection) | 32 |
| Training epochs per rollout | 2 |
| Minibatch count per epoch | 32 |
| *Exploration (Epsilon-Greedy)* | |
| $\epsilon$ start | 1.0 |
| $\epsilon$ finish | 0.001 |
| $\epsilon$ anneal period | 10% of training updates ($5 \times 10^6$ steps) |
| *Learning Parameters* | |
| Optimizer | RAdam |
| Discount factor ($\gamma$) | 0.99 |
| Lambda for TD($\lambda$) returns ($\lambda$) | 0.65 |
| Linear learning rate decay (Baseline & SCORER) | True (over 100% of training) |
| Target network | N/A (uses $\lambda$-returns) |
| Q-network learning rate ($\alpha_\theta$) | $2.5 \times 10^{-4}$ |
| Q-network max gradient norm | 10.0 |

Table 9: SCORER-Specific Hyperparameters for PQN (Atari environment defaults).

| Parameter | Value |
|---|---|
| *Leader-Follower Architecture* | |
| Optimizer | RAdam |
| Leader (Q-Network) learning rate ($\alpha_\theta$) | $2.5 \times 10^{-4}$ |
| Follower (Perception) learning rate ($\alpha_\phi$) | $5 \times 10^{-4}$ |
| Max gradient norm | 10.0 |
| Latent dimension | 256 |

Table 10: General Training Hyperparameters for SCORER R2D2.

| Parameter | Value |
|---|---|
| *Training Configuration* | |
| Number of parallel environments | 10 |
| Total timesteps | $1 \times 10^6$ |
| Learning starts (env steps) | 25,000 |
| *Prioritized Trajectory Replay Buffer* | |
| Buffer size (transitions) | $1 \times 10^5$ |
| Min buffer size for sampling | 5,000 |
| Batch size (sequences) | 32 |
| Sampled sequence length | 100 |
| Burn-in length | 50 |
| N-step returns | 10 |
| PER alpha ($\alpha$) | 0.6 |
| PER beta start ($\beta_0$) | 0.4 |
| PER beta end ($\beta_T$) | 1.0 |
| *Exploration (Epsilon-Greedy)* | |
| $\epsilon$ start | 1.0 |
| $\epsilon$ finish | 0.01 |
| $\epsilon$ anneal time (env steps) | $5 \times 10^5$ |
| *Learning Parameters* | |
| Optimizer | Adam |
| Discount factor ($\gamma$) | 0.99 |
| Linear learning rate decay (Baseline & SCORER) | True |
| Target network update interval (env steps) | 5,000 |
| Update interval (env steps) | 16 |
| Soft update parameter ($\tau$) | 1.0 |
| Q-network learning rate ($\alpha_\theta$) | $1 \times 10^{-4}$ |
| Max gradient norm | 0.5 |

Table 11: SCORER-Specific Hyperparameters for R2D2.

| Parameter | Value |
|---|---|
| *Leader-Follower Architecture* | |
| Optimizer | Adam |
| Leader (Control) learning rate ($\alpha_\theta$) | $1 \times 10^{-4}$ |
| Follower (Perception) learning rate ($\alpha_\phi$) | $3 \times 10^{-4}$ |
| Max gradient norm (Leader & Follower) | 0.5 |

## C.1 Network Architectures

All agents within the DQN family (DQN, DDQN, Dueling DQN, and Dueling DDQN) are built from a shared architectural template. The monolithic baseline for each variant is constructed first, and its corresponding SCORER version is created by splitting this same architecture to maintain parity in the number of layers and parameters. Table 12 details this base architecture.

Table 12: Base network architecture for the SCORER DQN agent family. `Linear(in, out)` denotes a fully connected layer and `L1Norm` denotes a layer that normalizes its input to have a unit L1 norm. Key dimensions are environment-dependent. The `activation` function is applied after hidden layers, but not on the final Q-Network output.

---

**SCORER DQN Base Architecture**

---

**Hyperparameters:**
`state_dim` = Dimension of observation vector
`action_dim` = Number of discrete actions
`latent_dim` = 64 (for Control tasks), 128 (for MinAtar)
`q_hidden_dim` = 64 (for Control tasks), 128 (for MinAtar)
`activation` = Tanh (for Control tasks), ReLU (for MinAtar)

---

**Perception Network (Follower, $f_\phi$)**
▷ `Encodes raw state into a latent representation z.`
*If MinAtar environment:*
`  input_state = L1Norm(state)`
`Else:`
`  input_state = state`
`p_l0 = Linear(state_dim, latent_dim)`

**Perception Network Forward Pass:**
`h0 = activation(p_l0(input_state))`
`z = h0   (For Control tasks)`
*If MinAtar environment:*
`  p_l1 = Linear(latent_dim, latent_dim)`
`  h1 = activation(p_l1(h0))`
`  z = L1Norm(h1)   (Overrides z)`

---

**Q Network (Leader, $Q_\theta$)**
▷ `Predicts Q-values from the latent representation z.`
`q_l0 = Linear(latent_dim, q_hidden_dim)`
`q_l1 = Linear(q_hidden_dim, action_dim)`

**Q Network Forward Pass:**
`input = z`
`h0 = activation(q_l0(input))`
`q_values = q_l1(h0)`

---

**Architectural Variants**  The architecture described in Table 12 serves as the foundation for all agents in the DQN family.

- **DQN and DDQN:** The monolithic baselines for DQN and DDQN consist of a single network formed by composing the layers described above (e.g., for MinAtar, a four-layer MLP). The SCORER variants are created by splitting this architecture exactly as shown above.

- **Dueling DQN and Dueling DDQN:** For these agents, the Perception Network ($f_\phi$) remains identical. The Q-Network ($Q_\theta$) is modified to implement the dueling architecture (Wang et al., 2016). Specifically, the 'h0' feature vector from the Q Network's forward pass is fed into two separate heads: a state-value head, $V(z) = $ Linear(q_hidden_dim, 1), and an advantage head, $A(z, a) = $ Linear(q_hidden_dim, action_dim). The final Q-values are then combined using the standard aggregation method: $Q(z, a) = V(z) + (A(z, a) - \frac{1}{|\mathcal{A}|} \sum_{a'} A(z, a'))$.

Table 13: Base network architecture for the SCORER PQN. Linear(in, out) denotes a fully connected layer, Conv(k, c, s) a convolutional layer, and Norm denotes LayerNorm. Key dimensions are environment-dependent. The activation function is applied in hidden layers.

---

**SCORER PQN Base Architecture**

---

**Hyperparameters:**
state_dim = Dimension of observation vector (or image shape for Atari)
action_dim = Number of discrete actions
latent_dim = 64 (Control), 128 (MinAtar), 256 (Atari)
q_hidden_dim = 32 (Control), 64 (MinAtar), 256 (Atari)
activation = ReLU (for all tasks)
norm_type = LayerNorm (for all tasks)

---

**Perception Network (Follower, $f_\phi$)**
▷ Encodes raw state into a latent representation z.

*Option A: Vector Observation (Control, MinAtar)*
```
p_l0 = Linear(state_dim, latent_dim)
```
**Forward Pass:**
```
h0 = p_l0(state)
h1 = Norm(h0)
h2 = activation(h1)
z = Norm(h2)
```

*Option B: Pixel Observation (Atari)*
```
cnn_block = [Conv(8x8, 32, 4), Norm, Relu, Conv(4x4, 64, 2),
Norm, Relu, Conv(3x3, 64, 1), Norm, Relu]
p_l0 = Linear(cnn_out_dim, latent_dim)
```
**Forward Pass:**
```
features = cnn_block(state)
h0 = p_l0(features.flatten())
h1 = Norm(h0)
z = activation(h1)    (Atari ends with activation)
```

---

**Q Network (Leader, $Q_\theta$)**
▷ Predicts Q-values from the latent representation z.
▷ Contains q_num_layers hidden blocks.
```
q_l_hidden = Linear(latent_dim, q_hidden_dim)
q_l_out = Linear(q_hidden_dim, action_dim)
```

**Q Network Forward Pass (for `q_num_layers=1`):**
```
input = z
h0 = q_l_hidden(input)
h1 = Norm(h0)
h2 = activation(h1)
q_values = q_l_out(h2)
```

---

Table 14: Architecture for the SCORER R2D2. `Linear(in, out)` denotes a fully connected layer, `GRUCell(in, out)` denotes a Gated Recurrent Unit cell, and `L2Norm` denotes a layer that normalizes its input to have a unit L2 norm. The `activation` function is applied after hidden layers, but not on the final Control Network output.

---

**SCORER R2D2 Base Architecture**

---

**Hyperparameters:**
```
state_dim = Dimension of observation vector
action_dim = Number of discrete actions
embed_dim = 128
recurrent_dim = 128
q_hidden_dim = 128
activation = Tanh
```

---

**Perception Network (Follower, $f_\phi$)**
▷ Encodes state and updates the recurrent hidden state.
▷ Takes current state $state_t$ and previous hidden state $h_{t-1}$ as input.
```
p_l_embed = Linear(state_dim, embed_dim)
p_gru = GRUCell(embed_dim, recurrent_dim)
```

**Perception Network Forward Pass:**
```
input_state = L2Norm(state_t)
emb = p_l_embed(input_state)
emb_act = activation(emb)
```
$h_t$, `z_raw = p_gru(`$h_{t-1}$`, emb_act)` *(h_t is new hidden, z_raw is GRU output)*
`z = L2Norm(z_raw)` *(Representation passed to Control Network)*
▷ The new hidden state for the next step is `L2Norm(`$h_t$`)`.

---

**Control Network (Leader, $Q_\theta$)**
▷ Predicts Q-values from the latent representation z.
```
q_l_hidden = Linear(recurrent_dim, q_hidden_dim)
q_l_out = Linear(q_hidden_dim, action_dim)
```

**Control Network Forward Pass:**
```
input = z
h0 = q_l_hidden(input)
h1 = activation(h0)
q_values = q_l_out(h1)
```

---

# D  OBJECTIVE FUNCTIONS

This section provides the explicit formulations used by the SCORER framework. In our game, the Control Network ($Q_\theta$) acts as the Leader and the Perception Network ($f_\phi$) acts as the Follower.

## D.1  LEADER (CONTROL NETWORK) OBJECTIVE

As detailed in Section 4.1 ( equation 1), the control network $Q_\theta$ acts as the Stackelberg leader. Its objective is to learn an optimal action-value function by minimizing the Mean Squared Bellman Error (MSBE). Given a batch of transitions $B_{\text{leader}}$ and the follower's best-reponse representations $f_{\phi^*}$, the leader's objective is:

$$\mathcal{L}_{\text{leader}}(\theta, \phi^*) = \mathbb{E}_{(s,a,r,s',d) \sim B_{\text{leader}}} \left[ (Y - Q_\theta(f_{\phi^*}(s), a))^2 \right]. \tag{7}$$

The Bellman target $Y$ used in these objectives is defined in Section 3. In our two-timescale algorithm, the leader's update at step $k+1$ uses the follower's most recent parameters $\phi_{k+1}$ as a proxy for the ideal $\phi^*$, and the gradient is taken with respect to $\theta$ only.

## D.2  FOLLOWER (PERCEPTION NETWORK) OBJECTIVE

The follower's goal is to learn representations that support the Leader's learning process. In our ablation studies, we explored two objectives for the follower, both of which treat the leader's parameters $\theta$ as fixed via a stop-gradient operation. See equation 5, and Algorithm 1 Line 12.

### D.2.1  MINIMIZING FOLLOWER'S MEAN SQUARED BELLMAN ERROR (SCORER MSBE)

In this ablation variant, the follower's objective is identical to the leader's loss. Here, the follower learns representations that directly minimize the MSBE given the leader's committed weights:

$$\mathcal{L}_{\text{Follower}}^{\text{MSBE}}(\phi, \overline{\theta}) = \frac{1}{N} \sum_{j=1}^{N} \left[ (Y_j - Q_{\overline{\theta}}(f_\phi(s_j), a_j))^2 \right]. \tag{8}$$

The optimization is performed with respect to $\phi$.

### D.2.2  MINIMIZING BELLMAN ERROR VARIANCE

This is the main objective used in SCORER, designed to explicitly stabilize the leader's learning signal. The intuition behind is that by focusing on the consistency of the Bellman errors across a batch, the follower can learn representations that are robust to the noisy target inherent in TD learning. Let the Bellman error for a single transition be $\delta_j(\phi, \overline{\theta}) = Y_j - Q_{\overline{\theta}}(f_\phi(s_j), a_j)$. The follower's goal is to minimize the sample variance of these errors as:

$$\mathcal{L}_{\text{follower}}^{\text{Var}}(\phi, \overline{\theta}) = \frac{1}{N} \sum_{j=1}^{N} \left( \delta_j(\phi, \overline{\theta}) \right)^2 - \left( \frac{1}{N} \sum_{j=1}^{N} \delta_j(\phi, \overline{\theta}) \right)^2. \tag{9}$$

This formulation is the standard sample variance (Casella & Berger, 2024), which uses all $N$ samples from the batch for both terms, and is algebraically equivalent to $\frac{1}{N} \sum_{j=1}^{N} (\delta_j - \bar{\delta})^2$ where $\bar{\delta}$ denotes the sample mean.

This objective turns the follower into an active stabilizing agent. Because the leader's performance is evaluated on an independent sample batch of data ($B_{\text{leader}}$), the follower is asked to learn representations that create a generally stable learning signal, instead of overfitting the specifics of its own sampled batch ($B_{\text{follower}}$). The follower must find representations that make the leader's value predictions more uniform and predictable across the entire data distribution, truly dividing the task of stabilization from the immediate task of performance maximization. Our ablation studies (Section 5.5) empirically validate that this variance-minimization objective outperforms using MSBE for the SCORER's follower objective.

**Stochasticity and the Variance Objective.** A well-known theoretical challenge in minimizing Bellman residual errors is the need for unbiased gradients, which typically requires two independent samples of the next state (Baird, 1995). SCORER addresses this by minimizing the sample variance calculated over a batch as it treats the batch statistics as a deterministic objective, ensuring that the gradients are exact with respect to the sampled data.

In stochastic environments, the total variance of the Bellman error naturally includes irreducible environmental noise. Yet, this total variance decomposes into aleatoric variance (environment stochasticity) and epistemic variance (prediction inconsistency):

$$\text{Var}_{\text{total}}[\delta] \approx \text{Var}_{\text{aleatoric}} + \text{Var}_{\text{epistemic}} \tag{10}$$

Because the perception network cannot alter the environment's inherent noise ($\text{Var}_{\text{aleatoric}}$), reducing the sample variance targets the reduction of $\text{Var}_{\text{epistemic}}$. This way, we push the follower to learn representations that make the value function errors as consistent as possible across the batch, simplifying the learning signal for the leader. Our results on the Stochastic Deep Sea environment (Section 5.4) confirm that this objective successfully stabilizes learning even in the presence of significant environmental noise.

## E MOTIVATION FOR STACKELBERG ROLES

Assigning the role of the leader to the Control network ($Q_\theta$) and setting Perception ($f_\phi$) as the follower is grounded in the structural properties of the corresponding bi-level optimization. The ultimate goal of $Q$-learning is to identify the $Q$-function that minimizes MSBE as we discussed in Sec. 3. Comparing the optimization problems faced by the control and perception networks in equation 1 and equation 2, it is a natural setting to have equation 1 as the *outer problem* (sometimes also known as the upper-level problem in the literature, see, e.g. Zhang et al. (2024)) and equation 2 as the *inner (lower-level) problem* in the bi-level optimization connecting them. The leader-follower roles for the two networks in the Stackelberg game thus in turn reflect this relationship.

## F STATISTICAL SIGNIFICANCE CALCULATIONS

Throughout this work, reported performance metrics aim to provide a robust understanding of algorithm behavior across multiple independent trials. This appendix details the methods used for calculating and presenting these statistics.

### F.1 IQM AND CONFIDENCE INTERVALS

The learning curves represent the IQM (Interquartile Mean) of the total reward, averaged over independent runs (seeds). Following Agarwal et al. (2021b), the IQM statistic is a recommended evaluation metric in deep RL as it has the same properties as a 25% trimmed mean. The bottom 25% and top 25% of run scores are removed at each timestep before the mean is computed, which mitigates the effect of outliers without losing as much statistical power as the median.

The shaded areas represent 95% confidence intervals (CIs) estimated using the percentile bootstrap method (Agarwal et al., 2021b). We estimate the sampling distribution of the IQM by generating $B = 2000$ bootstrap samples that resample the seeds with replacement, from which the 95% CI is given by the 2.5th and 97.5th percentiles of the bootstrap distribution, without assuming normality. The curves are smoothed by a rolling mean over 50 timesteps, which is applied to both the IQM point estimates and the CI boundaries.

**IQM for final performance tables** For reporting final performance in tables, we first compute the mean return over the last 10% of training for each seed, which results in a more stable *final score* per run. We then compute the IQM over these final scores from all seeds using the 25% trimmed mean, and CIs are estimated via percentile bootstrap with $B = 2000$ resamples. To test for statistical significance when highlighting values in the tables, we perform a bootstrap difference test: we compute the distribution of IQM differences between the best-performing variant and each

competitor, and count the number of bootstrap samples in which the competitor matches or exceeds the best. If this proportion is less than 5%, we consider the difference to be statistically significant ($p < 0.05$).

**Normalized IQM Return**   For comparative analysis across environments with different reward scales, we present normalized IQM returns in some figures. The normalization is performed per environment by dividing all IQM values by the maximum IQM value reached across all methods in that specific environment. This helps us to develop a meaningful comparison of relative performance improvements across tasks while preserving the temporal dynamics of learning.

## F.2   SPEED COMPARISON

The *Speed Comparison* column in Table 1 reports the relative wall-clock time required to complete a full training run. For each base algorithm (e.g., DQN), the runtime of its standard monolithic version is normalized to 1.00x. The runtime of the corresponding SCORER variant is then reported as a multiple of this baseline. A value of 0.99x indicates that the SCORER variant was 1% faster than its baseline, whereas a value of 1.01% would indicate it was 1% slower.

## F.3   TIME-TO-THRESHOLD (TTT) CALCULATION

To provide a measure of sample efficiency over the MiniGrid experiments, we use a Time-to-Threshold (TTT) analysis. TTT quantifies the number of environment steps required for an agent to reliably fulfill the task, or at least reach a high level of performance. Our calculation follows a two-pass process to ensure fair comparisons across environments with different reward scales and performance ceilings.

### F.3.1   PERFORMANCE THRESHOLDS

Initially, for each evaluation environment, we determine the maximum asymptotic performance achieved across all runs of all methods. This sets an empirical "best-case" performance for the task. The performance threshold for each environment is then set to 99% of this maximum value.

### F.3.2   CALCULATING TTT AND SUCCESS RATE

In the second pass, for each individual seed of each method, we identify the first timestep at which the agent's episodic return meets or exceeds the calculated 99% performance threshold. This timestep is recorded as the TTT for that run. If a run fails to reach the threshold within the maximum allowed timesteps for that environment, its TTT is considered undefined. The final TTT reported in Table 4 is the mean over all successful runs. To capture the reliability of each method, we also present the Success Rate (SR), which is the percentage of the 30 independent runs that successfully reached the performance threshold.

## F.4   STATISTICAL SIGNIFICANCE TESTING

In Table 1, we highlight statistically significant performance differences within each algorithm family (e.g. DQN vs. SCORER-DQN) per environment. We first identify the variant with the highest IQM within each group. We then use a bootstrap difference test to compare each of the other variants to this top performer: for each of the $B = 2000$ bootstrap resamples, we compute the IQM of both variants and record the difference. If less than 5% of bootstrap samples have the competitor matching or exceeding the top performer, we deem this difference to be statistically significant ($p < 0.05$). All variants that are *not* significantly worse than the best are highlighted in green.

## G   SAMPLE EFFICIENCY RESULTS

To complement the learning curves presented in Section 5, this section provides a detailed breakdown of the sample efficiency for SCORER compared to the baseline agents across four MinAtar environments. Tables 15 through 18 report the mean number of training steps (in millions, $\pm$ 95% CI) required for an agent to reach pre-defined reward thresholds, averaged over 30 seeds. For each

experimental condition, a run is considered to have reached a threshold at the first time step where its individual performance curve meets or exceeds the target value. The tables report the statistics for all seeds that successfully reached the threshold. An "N/A" indicates that the Interquartile Mean (IQM) of the agent's runs failed to reach the threshold within $10^8$ steps. The highlighted values indicates the variant that achieved the given threshold in the fewest timesteps.

Across the board, the data shows the substantial impact of the SCORER framework on learning speed. The effect is most pronounced in Breakout(Table 15). Here, the SCORER agents are an order of magnitude more sample-efficient than their monolithic baselines.

Table 15: Sample efficiency (time steps in millions to reach reward thresholds) for Breakout-MinAtar.

| Model | Variant | Threshold 17 | Threshold 20 | Threshold 25 |
|---|---|---|---|---|
| DQN | Baseline | $39.1 \pm 7.6$ | N/A | N/A |
| | SCORER | $3.4 \pm 0.2$ | $4.8 \pm 0.5$ | $8.2 \pm 0.9$ |
| DDQN | Baseline | $42.4 \pm 6.6$ | $43.8 \pm 5.0$ | $62.3 \pm 6.4$ |
| | SCORER | $4.2 \pm 0.4$ | $5.9 \pm 0.8$ | $12.7 \pm 3.6$ |
| DuelingDQN | Baseline | $43.6 \pm 8.4$ | $50.8 \pm 9.7$ | $72.0 \pm 9.1$ |
| | SCORER | $4.0 \pm 0.4$ | $5.3 \pm 0.6$ | $13.5 \pm 2.7$ |
| DuelingDDQN | Baseline | $27.6 \pm 6.5$ | $33.7 \pm 5.8$ | $64.2 \pm 5.2$ |
| | SCORER | $4.2 \pm 0.4$ | $5.6 \pm 0.6$ | $17.8 \pm 7.9$ |

In Asterix and SpaceInvaders Table 16 and Table 17), SCORER consistently reduces the number of samples required to reach performance thresholds, demonstrating robust improvements in sample efficiency across different base algorithms. The gains are particularly notable for higher thresholds.

In Freeway (Table 18), all agents converge rapidly. Even so, the SCORER variants consistently reach all performance thresholds in approximately half the time of their baseline counterparts.

Table 16: Sample efficiency (time steps in millions to reach reward thresholds) for Asterix-MinAtar.

| Model | Variant | Threshold 35 | Threshold 40 |
|---|---|---|---|
| DQN | Baseline | $22.8 \pm 0.6$ | $27.3 \pm 1.1$ |
| | SCORER | $14.6 \pm 2.2$ | $17.3 \pm 2.4$ |
| DDQN | Baseline | $23.3 \pm 0.9$ | $27.7 \pm 1.2$ |
| | SCORER | $14.8 \pm 1.0$ | $18.2 \pm 1.2$ |
| DuelingDQN | Baseline | $19.8 \pm 1.0$ | $22.6 \pm 1.0$ |
| | SCORER | $15.4 \pm 3.3$ | $18.3 \pm 3.4$ |
| DuelingDDQN | Baseline | $20.5 \pm 1.0$ | $25.1 \pm 1.8$ |
| | SCORER | $17.2 \pm 6.4$ | $16.6 \pm 3.2$ |

Table 17: Sample efficiency (time steps in millions to reach reward thresholds) for SpaceInvaders-MinAtar.

| Model | Variant | Threshold 75 | Threshold 100 | Threshold 110 |
|---|---|---|---|---|
| DQN | Baseline | $9.7 \pm 1.0$ | $20.6 \pm 2.7$ | $31.2 \pm 2.7$ |
| | SCORER | $9.1 \pm 0.5$ | $14.3 \pm 0.8$ | $17.8 \pm 1.7$ |
| DDQN | Baseline | $11.8 \pm 1.1$ | $24.7 \pm 2.7$ | $37.1 \pm 2.6$ |
| | SCORER | $11.1 \pm 0.6$ | $18.6 \pm 2.5$ | $22.9 \pm 3.4$ |
| DuelingDQN | Baseline | $10.0 \pm 0.9$ | $23.4 \pm 4.1$ | $33.2 \pm 5.7$ |
| | SCORER | $17.1 \pm 0.7$ | $26.0 \pm 1.8$ | $32.0 \pm 2.5$ |
| DuelingDDQN | Baseline | $9.9 \pm 0.7$ | $16.6 \pm 1.4$ | $26.9 \pm 3.1$ |
| | SCORER | $15.4 \pm 0.7$ | $23.8 \pm 1.8$ | $30.2 \pm 3.4$ |

Table 18: Sample efficiency (time steps in millions to reach reward thresholds) for Freeway-MinAtar.

| Model | Variant | Threshold 30 | Threshold 40 | Threshold 50 |
|---|---|---|---|---|
| DQN | Baseline | $3.8 \pm 0.2$ | $6.0 \pm 0.3$ | $10.1 \pm 0.5$ |
| | SCORER | $1.3 \pm 0.1$ | $2.3 \pm 0.1$ | $4.5 \pm 0.1$ |
| DDQN | Baseline | $3.8 \pm 0.3$ | $5.9 \pm 0.3$ | $9.6 \pm 0.4$ |
| | SCORER | $1.4 \pm 0.1$ | $2.3 \pm 0.1$ | $4.5 \pm 0.2$ |
| DuelingDQN | Baseline | $4.7 \pm 0.4$ | $6.1 \pm 0.3$ | $10.0 \pm 0.4$ |
| | SCORER | $1.8 \pm 0.2$ | $2.7 \pm 0.2$ | $5.0 \pm 0.4$ |
| DuelingDDQN | Baseline | $4.6 \pm 0.5$ | $6.8 \pm 0.7$ | $11.5 \pm 1.1$ |
| | SCORER | $1.6 \pm 0.2$ | $2.6 \pm 0.2$ | $4.9 \pm 0.4$ |

## H  CLASSIC CONTROL RESULTS

To further assess the general applicability of the SCORER framework beyond the MinAtar suite and MiniGrid, we conducted experiments on two classic control environments from OpenAI Gym (Brockman et al., 2016) using Gymnax (Lange, 2022) for JAX compatibility: For these tasks, simpler Multi-Layer Perceptron architectures were used for both the perception and control networks, with details provided in Appendix C.

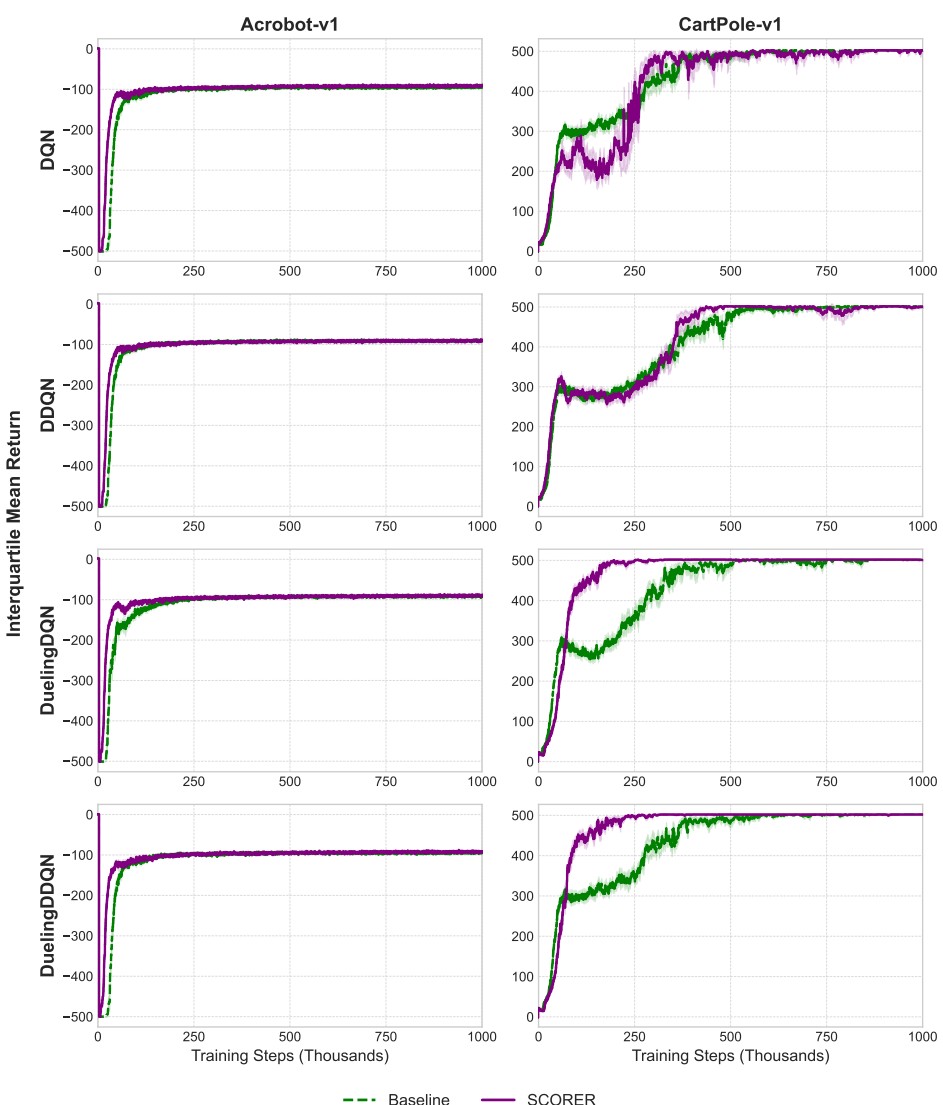

Figure 7: Learning curves on classic control environments (CartPole-v1, Acrobot-v1). Each row corresponds to a base algorithm (DQN, DDQN, DuelingDQN, DuelingDDQN), and each column to an environment. Curves show IQM return over 30 seeds; shaded regions represent 95% confidence intervals.

# I   LEARNING CURVES ON MINIGRID ENVIRONMENTS

This section provides the full learning curves that support the time-to-threshold analysis presented in Table 4. We evaluate the performance of SCORER when applied to R2D2, a recurrent baseline for partially observable environments, across a suite of MiniGrid tasks.

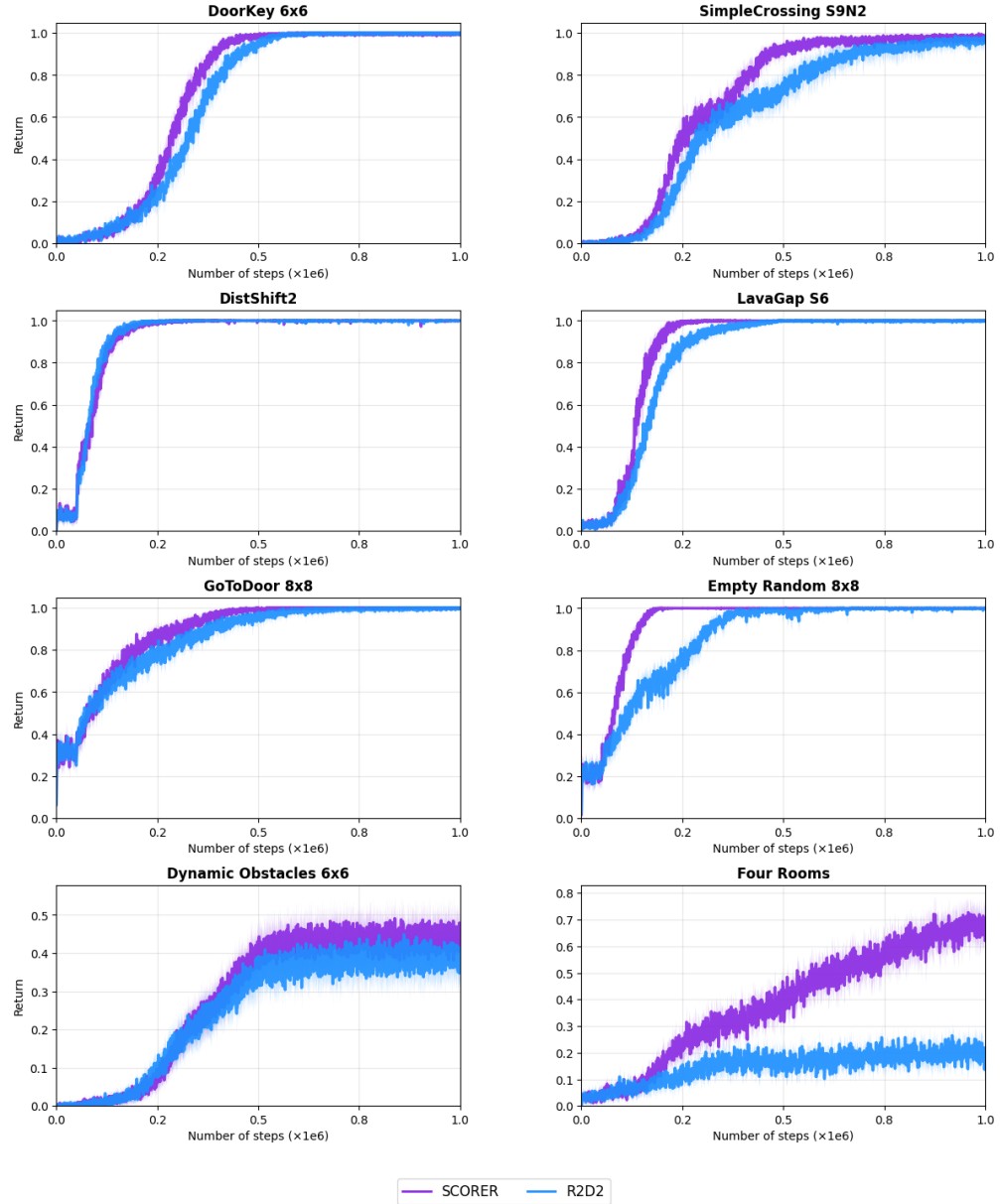

Figure 8: Learning curves for SCORER R2D2 SCORER versus the Vanilla R2D2 on eight Minigrid environments, averaged over 30 seeds. Shaded regions represent a 95% confidence interval. The plots highlight SCORER's consistent improvement in sample efficiency and its ability to solve challenging exploration tasks like Four Rooms where the baseline fails.

The learning curves in Figure 8 visually confirm Table 4's results. The SCORER-enhanced agent demonstrates a steeper learning curve in most environments, indicating superior sample efficiency. The most significant result is observed in the Four Rooms environment, a classic hard-exploration task. Here, the baseline R2D2 agent performs poorly, compared to SCORER R2D2, which can solve the task, reaching a high success rate.

## J    DEEP SEA RESULTS

Even in stochastic MDPs, minimizing Bellman error variance remains beneficial. The total variance decomposes as $\mathrm{Var}[\delta] = \mathrm{Var}_{\text{aleatoric}} + \mathrm{Var}_{\text{epistemic}}$, where aleatoric variance comes from environment stochasticity (non-reducible) and epistemic variance comes from value function error (reducible). The follower learns representations that minimize the total variance, which indirectly reduces epistemic variance by making the value function more consistent across the batch. We validate this using stochastic Deep Sea environment (depth 10) the suite's baseline (Bootstrapped DQN) (Osband et al., 2020; 2016). Deep sea is a challenging exploration task available in both deterministic and stochastic variants.

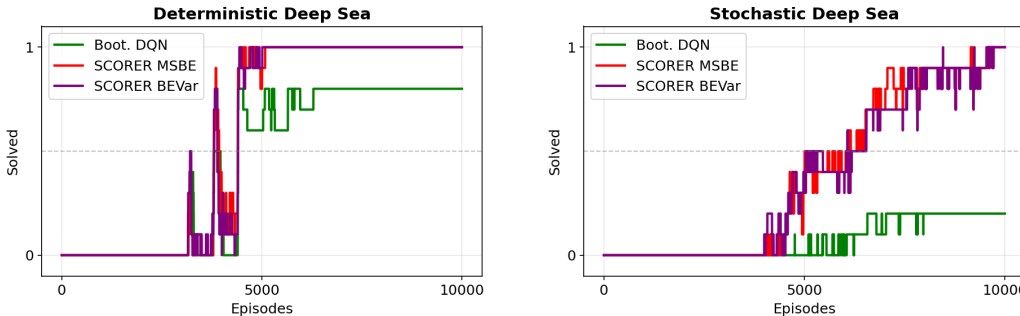

Figure 9: **SCORER Performance on Deterministic and Stochastic Deep Sea (Depth 10). (Left)** Deterministic Deep Sea: Both SCORER variants converge substantially faster than baseline Bootstrapped DQN, with near-complete solved rates by episode 5000. **(Right)** Stochastic Deep Sea: SCORER succeeds despite increased environment noise, achieving near-complete solved rates while the baseline reaches around 20%. The variance-minimization objective remains effective even in the presence of irreducible stochasticity.

## K    DETAILED ABLATION STUDY RESULTS

This section provides detailed, per-environment learning curves for the ablation studies summarized in Section 5.5.

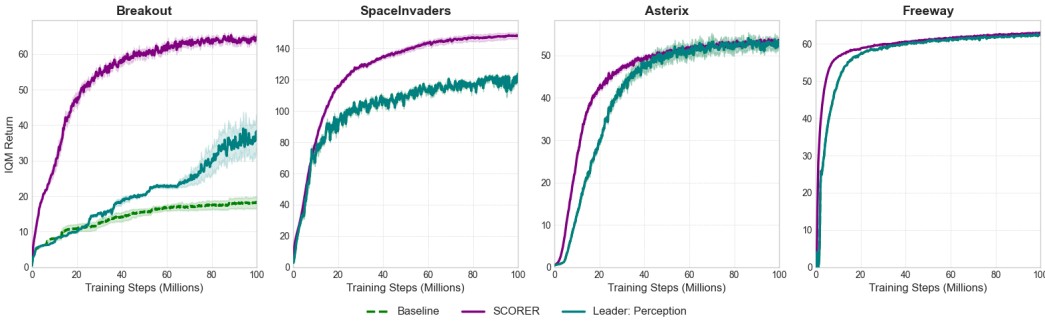

Figure 10: Per-environment results for the Stackelberg role assignment ablation. The standard SCORER configuration (Control as Leader, purple) is compared against the monolithic DQN baseline (green) and an inverted hierarchy where Perception acts as the Leader (teal).

**Stackelberg Role Assignment.**    Figure 10 details the results of our role-swapping experiment. The plots confirm that the standard SCORER configuration, where the control network acts as the leader, consistently and substantially outperforms both the baseline and the inverted hierarchy across all tested environments. The performance collapse observed when perception is assigned the leader role is particularly pronounced in complex environments like Breakout and SpaceInvaders, providing strong evidence that for stable learning to occur, the value function must lead the representation learning process.

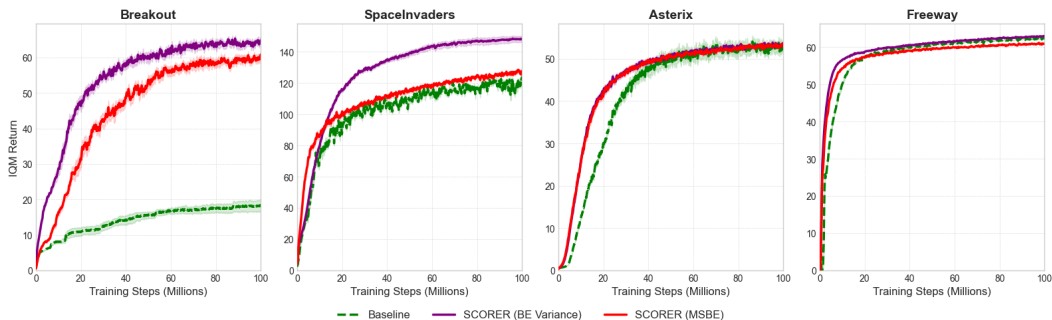

Figure 11: Perception Objective

**Follower's Objective.** Figure 11 presents a per-environment breakdown of the follower's objective ablation. The results reinforce the conclusion from our main analysis: the Bellman Error Variance objective is critical for SCORER's performance. In every environment, the BE Variance follower achieves the highest final performance and demonstrates the best sample efficiency. While the MSBE follower offers a clear improvement over the baseline, particularly in Breakout, the explicit stabilization provided by the variance objective unlocks a significantly higher level of performance, highlighting its role in enabling robust co-adaptation.

**Learning Rate Sensitivity.** We study SCORER's sensitivity to the timescale separation by testing on different learning rates for the follower ($\alpha_\phi$) while fixing the leader's rate ($\alpha_\theta = 1 \times 10^{-4}$). Figure 12 compares follower rates from $2 \times 10^{-4}$ to $1 \times 10^{-3}$ (ratios of 2:1 to 10:1). The results show that SCORER is not hypersensitive to the exact ratio; nonetheless, performance drops noticeably at the lowest rate ($2 \times 10^{-4}$), confirming the necessity for the follower to adapt sufficiently fast relative to the leader to compute an effective best response.

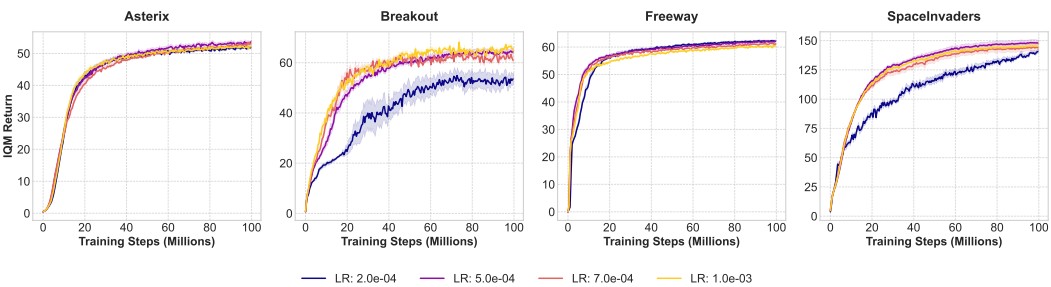

Figure 12: Performance across MinAtar environments with varying follower learning rates ($\alpha_\phi$), given a fixed leader rate

**Hierarchical vs. Synchronous Ablation** Figure 13 shows per-environment learning curves comparing all ablation variants. The monolithic per-layer LR baseline (encoders: $5 \times 10^{-4}$, Q-heads: $1 \times 10^{-4}$) outperforms the baseline in Breakout and SpaceInvaders, confirming that utilizing faster representation updates is of benefit. However, SCORER still surpasses this much simpler alternative.

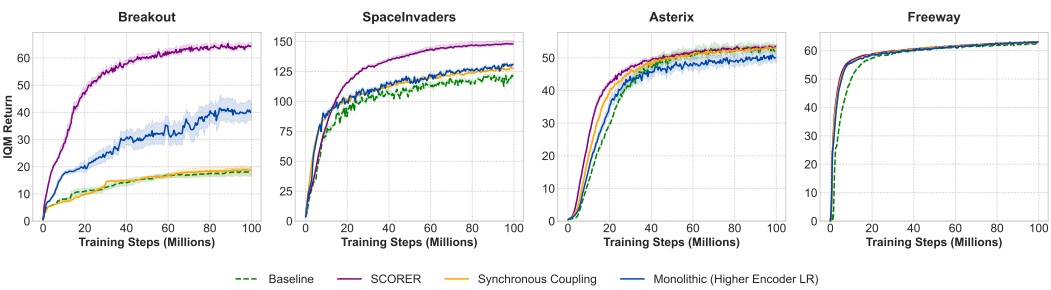

Figure 13: Team Coupling Study across MinAtar environments

## L    LEARNING DYNAMICS AND REPRESENTATION ANALYSIS

To gain further understanding into SCORER's behavior, we analyze dynamics of representation rank and parameter norm on Breakout-MinAtar, motivated by recent diagnostic efforts on the causes of deep RL pathologies (Kumar et al., 2021).

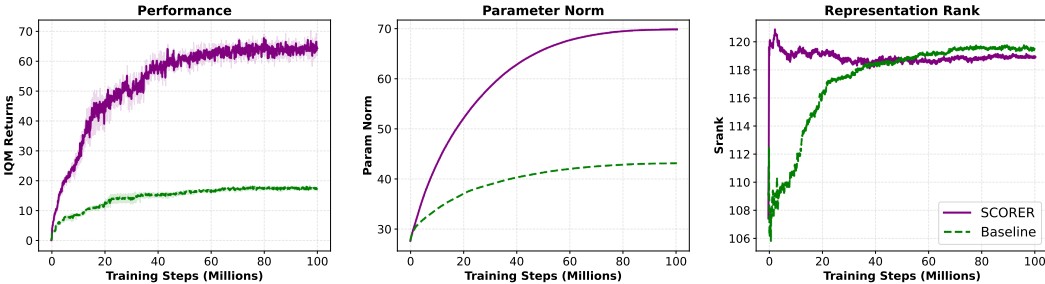

Figure 14: Learning dynamics on Breakout-MinAtar. (**Left**) Returns over training, showing SCORER's sample efficiency advantage and higher performance. (**Center**) $L_2$ parameter norm of all network weights; both methods eventually stabilize, but SCORER achieves a larger value. (**Right**) Srank; SCORER maintains near-maximal rank throughout, whereas the baseline only recovers a comparable value after ∼40M steps.

**Representation Rank.** We track the effective rank of penultimate activations using Srank (Kumar et al., 2021), which is an estimate of the minimal number of singular value components required to account for 99% of their cumulative sum. Figure 14(**Right**) shows that the difference is in the *when* of high-rank representations, rather than the *whether*. Baseline DQN starts at a lower effective rank of around 107 (out of 128 dimensions), and takes about 40 million steps to approach near-maximal values around 119. SCORER reaches a high effective rank very early in training and continues to hover around that level, though with a slight dip during initial exploration. The period of maximal rank divergence (0–40M steps) coincides precisely with the phase where SCORER demonstrates its largest performance advantage (Figure 14, **Left**). While our experiments cannot establish causality, this temporal correspondence is consistent with recent findings that representation collapse (the failure to utilize available representational capacity) is a primary cause of sample inefficiency in deep Q-learning (Kumar et al., 2021; Lyle et al., 2022). SCORER's game-theoretic structure appears to provide a natural regularization that maintains high-rank representations throughout training.

**Parameter Norm.** Figure 14(**Center**) shows the $L_2$ norm of all network parameters throughout training. SCORER exhibits faster parameter norm growth than the baseline, with both methods stabilizing in the latter half of training.

## M    THEORETICAL FOUNDATIONS OF SCORER

In this section, we discuss optimization (especially bilevel optimization) related issues in the SCORER framework. Since the results are of independent interest for a large family of bi-level optimization problems, we adapted a general formulation.

### M.1    BILEVEL OPTIMIZATION & TWO-TIMESCALE ALGORITHMS

The SCORER framework formulates the interaction between perception and control as a Stackelberg game, which we cast as a standard bilevel optimization problem:

$$\min_{\mathbf{x} \in \mathbb{R}^{d_x}} \ell(\mathbf{x}) := f(\mathbf{x}, \mathbf{y}^*(\mathbf{x})), \quad \text{s.t. } \mathbf{y}^*(\mathbf{x}) \in \arg \min_{\mathbf{y} \in \mathbb{R}^{d_y}} g(\mathbf{x}, \mathbf{y}). \tag{11}$$

**Mapping to SCORER.**    In the context of SCORER, this general formulation maps directly to the SCORER framework as:

- The outer (leader) variable $\mathbf{x}$ corresponds to the control network parameters $\theta$.

- The inner (follower) variable $\mathbf{y}$ corresponds to the perception network parameters $\phi$.
- The outer objective $f(\mathbf{x}, \mathbf{y})$ is the Mean Squared Bellman Error (MSBE) loss for the leader.
- The inner objective $g(\mathbf{x}, \mathbf{y})$ is the Bellman Error Variance loss for the follower.

The remainder of this section will proceed with the general $(\mathbf{x}, \mathbf{y})$ notation to align with the standard optimization literature.

Two-Timescale Stochastic Approximation (TTSA) algorithm proposed in Borkar (1997) updates the variable through the following iterative step,

$$\mathbf{y}^{k+1} = \mathbf{y}^k - \beta_k D_g^k, \tag{12}$$

$$\mathbf{x}^{k+1} = \mathbf{x}^k - \alpha_k D_f^k, \tag{13}$$

where two sequences of learning rates $\{\alpha_k\}$ and $\{\beta_k\}$ satisfying $\alpha_k / \beta_k \to 0$, as $k \to \infty$; $D_f^k$ and $D_g^k$ represent stochastic estimates of the gradients $\overline{\nabla}_x f(\mathbf{x}^k, \mathbf{y}^{k+1})$ and $\nabla_y g(\mathbf{x}^k, \mathbf{y}^k)$, respectively, with $\overline{\nabla}_x f(x, y) := \nabla_x f(x, y) - \nabla_{xy}^2 g(x, y) \nabla_{yy}^2 g(x, y)^{-1} \nabla_y f(x, y)$. The following conditions on the objective functions and their approximations are discussed in Hong et al. (2023), here, while we follow mostly their notations and basic arguments, detailed calculations are modified to fit our purposes.

**Assumption 1.** The outer function $f(\mathbf{x}, \mathbf{y})$ satisfies gradient Lipschitz conditions: for any $\mathbf{y}_1 \neq \mathbf{y}_2 \in \mathbb{R}^*$,

$$\frac{\|\nabla_x f(\mathbf{x}, \mathbf{y}_1) - \nabla_x f(\mathbf{x}, \mathbf{y}_2)\|}{\|\mathbf{y}_1 - \mathbf{y}_2\|} \leq L_{f_x}, \quad \text{uniformly in } \mathbf{x},$$

$$\frac{\|\nabla_y f(\mathbf{x}, \mathbf{y}_1) - \nabla_y f(\mathbf{x}, \mathbf{y}_2)\|}{\|\mathbf{y}_1 - \mathbf{y}_2\|} \leq L_{f_y}, \quad \text{uniformly in } \mathbf{x},$$

and for any $\mathbf{x}_1 \neq \mathbf{x}_2 \in \mathbb{R}^{d_x}$,

$$\frac{\|\nabla_y f(\mathbf{x}_1, \mathbf{y}) - \nabla_y f(\mathbf{x}_1, \mathbf{y})\|}{\|\mathbf{x}_1 - \mathbf{x}_2\|} \leq \bar{L}_{f_y}, \quad \text{uniformly in } \mathbf{y}.$$

Gradient bound condition: $\|\nabla_y f(\mathbf{x}, \mathbf{y})\| \leq C_{f_y}$, uniformly in $\mathbf{x}$ and $\mathbf{y}$.

**Assumption 2.** The inner function $g(x, y) \in C^2(\Omega \times \mathbb{R}^*)$,

$$\frac{\|\nabla_y y(\mathbf{x}, \mathbf{y}_1) - \nabla_y g(\mathbf{x}, \mathbf{y}_2)\|}{\|\mathbf{y}_1 - \mathbf{y}_2\|} \leq L_g, \quad \text{uniformly in } \mathbf{x},$$

$$\frac{\|\nabla_{xy}^2 g(\mathbf{x}, \mathbf{y}_1) - \nabla_{xy}^2 g(\mathbf{x}, \mathbf{y}_2)\|}{\|\mathbf{y}_1 - \mathbf{y}_2\|} \leq L_{g_{xy}}, \quad \text{uniformly in } \mathbf{x},$$

$$\frac{\|\nabla_{yy}^2 g(\mathbf{x}, \mathbf{y}_1) - \nabla_{yy}^2 g(\mathbf{x}, \mathbf{y}_2)\|}{\|\mathbf{y}_1 - \mathbf{y}_2\|} \leq L_{g_{yy}}, \quad \text{uniformly in } \mathbf{x},$$

$$\frac{\|\nabla_{xy}^2 g(\mathbf{x}_1, \mathbf{y}) - \nabla_{xy}^2 g(\mathbf{x}_2, \mathbf{y})\|}{\|\mathbf{x}_2 - \mathbf{x}_2\|} \leq \bar{L}_{g_{xy}}, \quad \text{uniformly in } \mathbf{x},$$

$$\frac{\|\nabla_{yy}^2 g(\mathbf{x}_1, \mathbf{y}) - \nabla_{yy}^2 g(\mathbf{x}_2, \mathbf{y})\|}{\|\mathbf{x}_2 - \mathbf{x}_2\|} \leq \bar{L}_{g_{yy}}, \quad \text{uniformly in } \mathbf{x},$$

Convexity: For any $x \in \mathbb{R}^{d_x}$, $g(x, \cdot)$ is strongly convex in $\mathbf{y}$ with modulus $\mu_g > 0$. Hessian boundedness: $\|\nabla_{xy}^2 g(\mathbf{x}, \mathbf{y})\| \leq C_{g_{xy}}$, uniformly in $\mathbf{x}$ and $\mathbf{y}$.

It is known that under these assumptions, we have

**Lemma M.1** (Lemma 2.2 from Ghadimi & Wang (2018)).

$$\|\overline{\nabla}_x f(x,y) - \nabla\ell(x)\| \leq L(y^*(x) - y\|, \quad \|y^*(x_1) - y^*(x_2)\| \leq L_y\|x_1 - x_2\|, \quad (14)$$

$$\|\nabla\ell(x_1) - \nabla\ell(x_2)\| = \|\nabla f(x_1, y^*(x_1)) - \nabla f(x_2, y^*(x_2))\| \leq L_f\|x_1 - x_2\|, \quad (15)$$

with

$$L := L_{f_x} + \frac{L_{f_y}C_{g_{xy}}}{\mu_g} + C_{f_y}\left(\frac{L_{g_{xy}}}{\mu_g} + \frac{L_{g_{yy}}C_{g_{xy}}}{\mu_g^2}\right),$$

$$L_f := L_{f_x} + \frac{(\bar{L}_{f_y} + L)C_{g_{xy}}}{\mu_g} + C_{f_y}\left(\frac{\bar{L}_{g_{xy}}}{\mu_g} + \frac{\bar{L}_{g_{yy}}C_{g_{xy}}}{\mu_g^2}\right), L_y := \frac{C_{g_{xy}}}{\mu_g}.$$

We also make assumptions on the random approximation of the gradient. The widely used assumptions in optimization literature, see e.g. Hong et al. (2023); Li et al. (2024); Pan et al. (2025), are usually in the following form

**Assumption 3.** There are two positive constants $\sigma_f$ and $\sigma_g$, and a nonincreasing sequence $\{b_k\}_{k\geq 0}$ such that,

$$\mathbb{E}[D_g^k|\mathcal{F}_k] = \nabla g(x^k, y^k), \quad \mathbb{E}[D_g^k|\mathcal{F}_k'] = \nabla g(x^k, y^{k+1}) + B_k, \|B_k\| \leq b_k, \quad (16)$$

$$\mathbb{E}[\|D_g^k - \nabla_y g(x^k, y^k)\|^2|\mathcal{F}_k] \leq \sigma_g^2[1 + \|\nabla_y g(x^k, y^k)\|^2], \quad (17)$$

$$\mathbb{E}[\|D_f^k - \overline{\nabla}_x f(x^k, y^{k+1}) - B_k\|^2|\mathcal{F}_k'] \leq \sigma_f. \quad (18)$$

In this paper, the function and derivative approximation are realized through deep neural networks. Therefore, it is desired to connect the constants $\sigma_f$, $\sigma_g$, and $\{b_k\}_{k\geq 0}$ to network parameters, such as their widths and depths, which can be adjusted to ensure the assumptions hold. This type of quantitative relation between the effectiveness of the approximation and network parameters has been carried out recently, see, e.g. Lu et al. (2021) and Belomestny et al. (2023). Summarizing their results, for neural networks of width $W$ and depth $D$, the following can be reasonably assumed.

**Assumption 4.** There are two positive constants $\sigma_f$ and $\sigma_g$, and a nonincreasing sequence $\{b_k\}_{k\geq 0}$ such that,

$$\mathbb{E}[D_g^k|\mathcal{F}_k] = \nabla g(x^k, y^k), \quad \mathbb{E}[D_g^k|\mathcal{F}_k'] = \nabla g(x^k, y^{k+1}) + B_k, \|B_k\| \leq b_k\mathcal{S}, \quad (19)$$

$$\mathbb{E}[\|D_g^k - \nabla_y g(x^k, y^k)\|^2|\mathcal{F}_k] \leq \sigma_g^2[1 + \|\nabla_y g(x^k, y^k)\|^2]\mathcal{S}^2, \quad (20)$$

$$\mathbb{E}[\|D_f^k - \nabla g(x^k, y^{k+1}) - B_k\|^2|\mathcal{F}_k'] \leq \sigma_f\mathcal{S}^2. \quad (21)$$

with $\mathcal{S} := W^{-2s/(d_x+d_y)}D^{-2s/(d_x+d_y)}$.

**Assumption 5.** For any fixed $\mathbf{x} \in \mathbb{R}^{d_x}$, the inner objective function $g(\mathbf{x}, \cdot)$ satisfies the Restricted Secant Inequality with respect to its minimizer $\mathbf{y}^*(\mathbf{x})$. That is, for all $\mathbf{y} \in \mathbb{R}^{d_y}$, we have:

$$\langle \nabla_{\mathbf{y}} g(\mathbf{x}, \mathbf{y}), \mathbf{y} - \mathbf{y}^*(\mathbf{x}) \rangle \geq \mu_g\|\mathbf{y} - \mathbf{y}^*(\mathbf{x})\|^2$$

where $\mu_g > 0$ is a constant.

## M.2 CONVERGENCE ANALYSIS OF THE ALGORITHMS FOR INNER OPTIMIZATION

The update for the inner optimization (SGD) takes the following form,

$$y^{k+1} = y^k - \beta_k D_g^k, \quad (22)$$

where $D_g^k$ is a random variable approximating $\nabla_y g(x^k, y^k)$.

The goal is to estimate $\mathbb{E}[\|y^{k+1} - y^*(x^k)\|^2|\mathcal{F}_k]$, for that, we have,

$$\mathbb{E}[\|y^{k+1} - y^*(x^k)\|^2|\mathcal{F}_k] = \mathbb{E}[\|y^k - y^*(x^k) - \beta_k D_g^k\|^2|\mathcal{F}_k]$$

$$= \mathbb{E}[\|y^k - y^*(x^k)\|^2 - 2\beta_k D_g^k[y^k - y^*(x^k)] + \beta_k^2\|D_g^k\|^2|\mathcal{F}_k].$$

From restricted secant inequality (RSI), we know that

$$\mathbb{E}[D_g^k[y^k - y^*(x^k)]|\mathcal{F}_k] \overset{ind}{=} \nabla_y g(x^k, y^k)[y^k - y^*(x^k)] \overset{RSI}{\geq} \mu_g\|y^k - y^*(x^k)\|^2.$$

Plug this back into the equation above, we have,

$$\mathbb{E}[\|y^{k+1} - y^*(x^k)\|^2 | \mathcal{F}_k] \le \mathbb{E}(1 - 2\beta_k)\|y^k - y^*(x^k)\|^2 + \beta_k^2 \mathbb{E}[\|D_g^k\|^2 | \mathcal{F}_k].$$

Now, let us examine the last term,

$$\begin{aligned}
\mathbb{E}[\|D_g^k\|^2 | \mathcal{F}_k] &= \mathbb{E}[\|D_g^k\|^2 - \|\nabla_y g(x^k, y^k)\|^2 | \mathcal{F}_k] + \|\nabla_y g(x^k, y^k)\|^2 \\
&\overset{(1)}{=} \mathbb{E}[\|D_g^k - \nabla_y g(x^k, y^k)\|^2 | \mathcal{F}_k] + \|\nabla_y g(x^k, y^k)\|^2 \\
&\overset{(2)}{\le} \sigma_g^2 \mathcal{S}^2 + \sigma_g^2 \mathcal{S}^2 \|\nabla_y g(x^k, y^k)\|^2 + \|\nabla_y g(x^k, y^k)\|^2 \\
&= \sigma_g^2 \mathcal{S}^2 + (1 + \sigma_g^2 \mathcal{S}^2)\|\nabla_y g(x^k, y^k)\|^2 \\
&\overset{(3)}{=} \sigma_g^2 \mathcal{S}^2 + (1 + \sigma_g^2 \mathcal{S}^2)\|\nabla_y g(x^k, y^k) - \nabla_y g(x^k, y^*(x^k))\|^2 \\
&\overset{(4)}{=} \sigma_g^2 \mathcal{S}^2 + (1 + \sigma_g^2 \mathcal{S}^2) L_g^2 \|y^k - y^*(x^k)\|^2,
\end{aligned}$$

where (1) is due to the fact that $\mathbb{E}[D_g^k\| = \nabla_y g(x^k, y^k)$; (2) is due to the assumption on variance of $D_g^k$ in Assumption 4; (3) is due to the fact that $y^*(x^k)$ is the stationary point, hence $\nabla_y g(x^k, y^*(x^k)) = 0$; and (4) is due to the gradient Lipschitz assumption.

Now, with the assumption that $\beta_k$ goes to zero, as $k \to \infty$, $\beta_k(1 + \sigma_g^2) \le \mu_k$ always holds for sufficiently large $k$. Hence, we have,

$$\mathbb{E}[\|y^{k+1} - y^*(x^k)\|^2 | \mathcal{F}_k] \le (1 - \beta_k)\|y^k - y^*(x^k)\|^2 + \beta_k^2 \sigma_g^2.$$

Next, to form a recursion, split $\|y^k - y^*(x^k)\|^2$ into $\|y^k - y^*(x^{k-1})\|^2 + \|y^*(x^{k-1}) - y^*(x^k)\|^2$. Thus,

$$\|y^*(x^{k-1}) - y^*(x^k)\|^2 \le L_y^2 \|x^k - x^{k-1}\|^2 = L_y^2 \|\alpha_{k-1} D_f^{k-1}\|^2 = \alpha_{k-1}^2 L_y^2 \|D_f^{k-1}\|^2,$$

where the first inequality follows from the Lipschitz continuity of $\mathbf{y}^*(\cdot)$ (Lemma M.1), and the equality follows directly from the definition of the unconstrained update rule in Equation equation 13. Therefore,

$$\mathbb{E}[\|y^{k+1} - y^*(x^k)\|^2 | \mathcal{F}_k] \le \mathbb{E}(1 - \beta_k)\|y^k - y^*(x^{k-1})\|^2 + \beta_k^2 \sigma_g^2 + a_{k-1}^2 L_y^2 \mathbb{E}[\|D_f^{k-1}\|^2 | \mathcal{F}_k].$$

Hence, we have,

$$\begin{aligned}
&\mathbb{E}[\|y^{k+1} - y^*(x^k)\|^2 | \mathcal{F}_k] \\
&\le \mathbb{E}(1 - \beta_k + a_{k-1}^2 L_y^2 (1 + \sigma_g^2 \mathcal{S}^2) L_g^2)\|y^k - y^*(x^{k-1})\|^2 + \beta_k^2 \sigma_g^2 + a_{k-1}^2 L_y^2 \sigma_g^2 \mathcal{S}^2.
\end{aligned}$$

As we know $\alpha_k/\beta_k \to 0$, we can see that $(1 - \beta_k + a_{k-1}^2 L_y^2 B)$ will be the uniform contraction factor, and $\beta_k^2 \sigma_g^2 + a_{k-1}^2 L_y^2 A$ is a correction term also tends to zero, therefore, $\mathbb{E}[\|y^{k+1} - y^*(x^k)\|^2$ diminished to zero, and the rate can also be quantified, especially in terms of the size of the neural networks.

## M.3 Convergence analysis of the algorithms for outer optimization

From the unconstrained update rule in equation 13, we have:

$$\|x^{k+1} - x^*\|^2 = \|x^k \alpha_k D_f^k - x^*\|^2 = \|x^k - x^*\|^2 - 2\alpha_k \langle D_f^k, x^k - x^* \rangle + \alpha_k^2 \|Df^k\|^2,$$

where $x^*$ denotes the global optimum of problem defined in equation 11. From the Assumption 4 on the random variable $D_f^k$, we can see that,

$$\begin{aligned}
\mathbb{E}[\langle D_f^k, x^k - x^* \rangle | \mathcal{F}_k] &= \langle \nabla_x f(x^k, y^{k+1}) + B_k, x^k - x^* \rangle \\
&= \langle \nabla \ell(x^k), x^k - x^* \rangle + \langle \nabla_x f(x^k, y^{k+1}) - \nabla \ell(x^k) + B_k, x^k - x^* \rangle.
\end{aligned}$$

Hence, we have,

$$\begin{aligned}
\mathbb{E}[\|x^{k+1} - x^*\|^2 | \mathcal{F}_k] &\le \|x^k - x^*\|^2 - 2\alpha_k \langle \nabla \ell(x^k), x^k - x^* \rangle + \alpha_k^2 \mathbb{E}\|Df^k\|^2 | \mathcal{F}_k] \\
&\quad - 2\alpha_k \langle \nabla_x f(x^k, y^{k+1}) - \nabla \ell(x^k) + B_k, x^k - x^* \rangle.
\end{aligned}$$

Restricted secant inequality implies that,

$$\langle\nabla\ell(x^k), x^k - x^*\rangle = \langle\nabla\ell(x^k) - \nabla\ell(x^*), x^k - x^*\rangle \geq \mu_\ell\|x^k - x^*\|^2.$$

We then have,

$$\mathbb{E}[\|x^{k+1} - x^*\|^2|\mathcal{F}_k] \leq (1 - 2\alpha_k\mu_\ell)\|x^k - x^*\|^2 - 2\alpha_k\langle\nabla_x f(x^k, y^{k+1}) - \nabla\ell(x^k) + B_k, x^k - x^*\rangle$$
$$+ \alpha_k^2\mathbb{E}[\|Df^k\|^2|\mathcal{F}_k]$$
$$\overset{(1)}{\leq} (1 - \alpha_k\mu_\ell)\|x^k - x^*\|^2 + \frac{\alpha_k}{\mu_\ell}\|\nabla_x f(x^k, y^{k+1}) - \nabla\ell(x^k) + B_k\|^2$$
$$+ \alpha_k^2\mathbb{E}[\|Df^k\|^2|\mathcal{F}_k]$$
$$\overset{(2)}{\leq} (1 - \alpha_k\mu_\ell)\|x^k - x^*\|^2 + \frac{2\alpha_k}{\mu_\ell}[L^2\|y^{k+1} - y^*(x^k)\|^2 + b_k\mathcal{S}]$$
$$+ \alpha_k^2\mathbb{E}[\|Df^k\|^2|\mathcal{F}_k],$$

where (1) is the result of completing a square, and (2) again follows from Lemma M.1. Hence, we can have uniformly bounded constants $\pi, \zeta > 0$ such that

$$\mathbb{E}[\|x^{k+1} - x^*\|^2|\mathcal{F}_k] \leq (1 - \alpha_k\mu_\ell)\|x^k - x^*\|^2 + \alpha_k\pi L^2\|y^{k+1} - y^*(x^k)\|^2 + \zeta\alpha_k^2.$$

Incorporating the above estimation into the inner and outer optimization, following a similar argument to Theorem 1 in Hong et al. (2023), we can reach the following conclusion.

**Theorem M.1.** Under Assumptions 1, 2 and 4, when $\alpha_k \leq c_0\beta_k^{3/2}$ and $\beta_k \leq c_1\alpha_k^{2/3}$ with constants $c_0, c_1 > 0$, the difference between the $k$-th step of the algorithm and the global optimum of problem defined in equation 11, $x^*$, can be estimated as,

$$\mathbb{E}[\|x^k - x^*\|^2] \leq \left\{\prod_{i=0}^{k-1}((1 - \alpha_i\mu_\ell)\left[\mathbb{E}[\|x^0 - x^*\|^2] + \pi\mathbb{E}[\|y^0 - y^*(x^0)\|^2]\right] + \zeta\alpha_{k-1}^{2/3}\right\}. \quad (23)$$

Similarly,

$$\mathbb{E}[\|y^k - y^*(x^{k-1})\|^2] \leq \sigma_g^2(\mathcal{S}^2 + 3)\left[\prod_{i=0}^{k-1}\left(1 - \frac{\beta_i\mu_g}{4}\right)\mathbb{E}[\|y^0 - y^*(x^0)\|^2] + \beta_{k-1}\right]. \quad (24)$$

The consequence of the theorem is that in the case of $\alpha_k, \beta_k \to 0$, as $k \to \infty$, we know that the two quantities on the left-hand side will diminish to zero, thus the convergence of the algorithm in the sense of mean square error.

