# OpenReview forum: "Stackelberg Coupling of Online Representation  Learning and Reinforcement Learning"
_ICLR.cc/2026/Conference — ICLR 2026 Poster_

### Official Review · Reviewer_b2sk · 2025-10-15

**Soundness:** 3
**Presentation:** 3
**Contribution:** 2
**Rating:** 4
**Confidence:** 3

**Summary:**

This paper introduces Stackelberg Coupled Representation and Reinforcement Learning, a novel value-based RL framework inspired by game theory. The key insight is to decouple representation and value learning into a hierarchical leader-follower game: the Q-function (control network) is the leader updated on a slow timescale, while the perception network (state encoder) is the follower learning rapidly to minimize Bellman error variance under the leader’s strategy. This two-timescale design aims to solve the instability issue (deadly triad) seen in monolithic off-policy deep Q-learning, improving sample efficiency and performance without increased model complexity. Experiments on MinAtar, Atari, and MiniGrid show consistent, substantial improvements over standard DQN and its variants.

**Strengths:**

1. The paper is well-motivated by the Stackelberg game theory to reformulate the RL optimization problem, separating learning dynamics between representation and value estimation.

2. The role assignments (control as leader, perception as follower) and the choice of Bellman error variance for the follower are both theoretically justified and empirically validated.

**Weaknesses:**

1. While computational cost is minimal, the framework introduces new algorithmic complexity (two-level optimization, choice of timescale schedules) that requires parameter tuning.

2. The method is centered entirely on discrete value-based methods. Extension to actor-critic or continuous-control algorithms is absent.

**Questions:**

1. Do you consider DQN and Dueling DQN as good baselines and the results naturally generalize to strong baselines such as Rainbow?

2. Can you add studies or discussions on actor-critic methods?

---

> ### Author Response · Authors · 2025-11-21
>
> We thank the reviewer for recognizing the theoretical motivation and the justification of our game-theoretic roles.
>
> # W1:
> We appreciate your concern regarding algorithmic complexity and parameter tuning, and we would like to highlight that **SCORER introduces exactly one new hyperparameter (the follower's learning rate $\alpha_\phi$)**, making the tuning burden minimal by design. SCORER adopts the entire tuned setup from each baseline (e.g., network size, gradient clipping, decay schedules, etc.). The only hyperparameter to additionally tune is $\alpha_\phi$, which we simply set to satisfy the timescale separation condition ($\alpha_\phi > \alpha_\theta$). **In our revised submission, we included an ablation for the Perception's learning rate (Appendix K, Figure 12)** showing that **SCORER is not hypersensitive to this hyperparameter** and performs exceptionally well, especially in 5:1 to 10:1 ratios, confirming that its benefits stem from the game-theoretical design.
>
> # Q2 & W2:
> We appreciate this suggestion, as we mentioned actor-critic extension as future work.
>
> ## Our Focus on Value-Based Methods
> We chose to focus on value-based RL because the deadly triad (function approximation plus bootstrapping plus off-policy learning,[Van Hesselt et al., 2018]) represents a well-characterized instability in this setting. Our stability experiments (Section 5.4, in our revision) directly test whether SCORER addresses this instability. The Atari benchmark and partially observable environments like MiniGrid challenges remain important testbeds where value-based methods are widely used.
>
> ## Actor-Critic as Future Work.
> Extending to actor-critic would demand careful consideration and exploration with the hierarchical structure and design. We agree this is an important direction and view our value-based results as establishing the core game-theoretic principle that could then be adapted to other RL paradigms.
>
> We recognize that the current scope focuses on discrete action spaces, and we appreciate the reviewer pointing out this limitation.
>
> # Q1:
> ## Baselines and Generalization to Rainbow
> Thank you for asking this question about our baselines. While DQN and variants are core methods, we wanted to emphasize that our experimental suite also encompasses more recent baselines and that we believe these experiments help to answer the generalization question.
>
> **PQN as a Modern Baseline:** First, we include PQN [Gallici et al., 2025] as an SOTA point of comparison. As we discuss in Section 5, PQN attains better performance on MinAtar, and we empirically tested on Atari and MinAtar, so it is a relevant baseline for the question of generalization to classical DQN variants.
>
> **R2D2 for Partial Observability:** Second, we experiment on R2D2 [Kapturowski et al., 2018], which bundles up several more advanced components compared to DQN, including a recurrent network, prioritized replay, and n-step returns (exact configurations are in Tables 10 and 14). We find consistent benefits on **MiniGrid (Section 5.3, Table 4)**, even on a challenging exploration task such as Four Rooms.
>
> **Connection to Rainbow:** With respect to Rainbow specifically, we note that our present experiments already include three of the six components (Double Q-learning, Dueling architectures, and Prioritized Replay via R2D2). The other Rainbow components would naturally fit with SCORER's game-theoretic structure and can be explored in future work.
>
> [1] Hado Van Hasselt, Yotam Doron, Florian Strub, Matteo Hessel, Nicolas Sonnerat, and Joseph Modayil. Deep reinforcement learning and the deadly triad. arXiv preprint arXiv:1812.02648, 2018
>
> [2] Matteo Gallici, Mattie Fellows, Benjamin Ellis, Bartomeu Pou, Ivan Masmitja, Jakob Nicolaus Foerster, and Mario Martin. Simplifying deep temporal difference learning. In The Thirteenth International Conference on Learning Representations, 2025
>
> [3] Steven Kapturowski, Georg Ostrovski, John Quan, Remi Munos, and Will Dabney. Recurrent experience replay in distributed reinforcement learning. In International conference on learning representations, 2018.

---

### Official Review · Reviewer_HaiV · 2025-10-27

**Soundness:** 3
**Presentation:** 3
**Contribution:** 3
**Rating:** 8
**Confidence:** 3

**Summary:**

This paper presents a new framing of value-based deep RL methods in terms of a two-player game between a representation player and a controller player. The motivation stems from the apparent instability that emerges in RL when a learner tries to simultaneously learn its representation, and its control policy. The core technical idea is to treat the learner's estimate of the Q-function as the leader, while a perception network is treated as the follower of a Stackelberg game. This framing yields "Stackelberg Coupled Representation and Reinforcement Learning" (SCORER), a simple value-based method for deep RL based on the principles of Stackelberg games. The game is solved using a two-level form of gradient descent, which yields a stable relationship between the representation and the value. Critically, this game only involves the exchange of the standard signals of the RL problem (based around MSBE) and does not allow incorporation of extraneous signals (such as intrinsic motivation, entropy bonuses, and so on). This methodological choice allows the work to focus purely on the game dynamics in a simple setting. The paper's primary evidence to support its central claims comes from a broad experimental study contrasting the performance of SCORER with standard value-based methods (DQN, DDQN, and so on) across a variety of standard tasks.

**Strengths:**

- This is a very clean paper. The scope is well defined, the question is well-posed, all of the needed ingredients are self-contained within the paper, and the solution (two-level optimization of a Stackelberg game) is tidy. In other words, the paper does exactly what it sets out to do in the intro.
- The experimental study is well designed, with appropriately chosen baselines, a variety of environments, and suitable ablations to explore various design choices (such as which player should be the leader vs. follower in Fig 4).
- The writing is clear and does a good job of motivating the various design choices. I especially appreciated the discussion around whether extraneous other signals should be incorporated into the study such as standard self-supervised objectives. I found the question, the main method, and the findings to be well-motivated and communicated effectively.
- The convergence result in the appendix adds a firm theoretical support to the approach.
- Lastly, the idea at the heart of the paper is relatively simple: if we decouple algorithmic components, we can treat them as players in a game, and solve the game.

**Weaknesses:**

- The general premise of mitigating challenges of deep RL through decomposing the monolithic policy network or value network into different components has been studied extensively. The proposal to treat a two-decomposition sub-system as engaged in a Stackelberg game is, to my knowledge, new, though I do believe a deeper understanding of why this specific decomposition is valuable would strengthen the work. As two concrete examples:

1. In Anand and Precup (2023), they propose to decompose the estimate of the value function into a "permanent" and "transient" component. These two components yields two separate networks, one for learning the permanent and one for the transient. They key difference between the two is that they are updated at different schedules. Then, the overall value function is determined by taking a combination of the two. I believe there is a heavy conceptual overlap in this approach and SCORER, and I wonder: if someone were to want to deploy a value-based method for RL, when should they use SCORER as approach to the approach of Anand and Precup?

2. Second, the Actor-Critic architecture is naturally decomposed into separate pieces to enhance stability. Indeed, a recent paper by Garcin et al (2025) has explicitly looked at the learned representations in each of the Actor and Critic, and found that the two networks tend to specialize their representation differently. First, I believe applying this kind of analysis to SCORER could be beneficial to shed further light on what specifically is being learned by the representation and policy in each of the ablations (and in the general setup). Second, it is so far again unclear what the precise benefit of this proposed decomposition is compared to something like Actor-Critic. I see this is noted in the discussion (applying SCORER to AC algorithms), though I believe this is an important point to understand about the space.

- Otherwise, no obvious weaknesses stand out. This is a clean paper with a tight focus that delivers on what it sets out to do.


Some typos and writing suggestions:

- Some repeated text at line 166 or so: "The defining characteristic is the leader’s ability to anticipate and influence the follower’s decision by committing to a strategy first. The defining characteristic is the leader’s ability to anticipate the follower’s best response and commit to a strategy that steers the game toward a favorable equilibrium.".

- Convergence analysis in Appendix J is really nice. If you can find space, I would love to see this in the main text, even it is just a colloquial theorem statement with the assumptions and proof in the appendix. On that note, I would suggest stating the convergence result as a mathematical theorem or proposition, even if it is in the appendix, to give you a chance to succinctly assert the result with all needed quantifiers and variables in one place.

References:
- Anand, N., & Precup, D. Prediction and control in continual reinforcement learning. NeurIPS 2023.
- Garcin, S., McInroe, T., Castro, P. S., Panangaden, P., Lucas, C. G., Abel, D., & Albrecht, S. V. Studying the Interplay Between the Actor and Critic Representations in Reinforcement Learning. ICLR 2025.

**Questions:**

My main question relating to my evaluation of the work is as follows:

MQ.1: For the purposes of understanding, how do you see SCORER in relation to other approaches to deep RL that decompose the monolith into a variety of subcomponents? Can you comment on the merits of framing this as perceiver/actor in a game, rather than something else (such as the Anand and Precup approach)?

Separately, I have a one other question (OQ.X), though it is less critical to my overall appraisal:

OQ.1: How do you foresee SCORER extending to another similar algorithmic templates, such as Actor-Critic, or even cases when value is not learned at all but perhaps just a policy?

---

> ### Author Response · Authors · 2025-11-23
>
> We are grateful to the reviewer for the constructive and encouraging review of our work. We are glad that the reviewer found the paper easy to follow and the reasoning behind the Stackelberg formulation and our experiments. Thanks for the insightful connections drawn to recent literature on decomposition in reinforcement learning, allowing us to further clarify the positioning of SCORER.
>
> **The Structural difference between SCORER and Anand & Precup (2023) (MQ.1)**
>
> We thank the reviewer for bringing this line of research to our attention. Indeed, we agree that both SCORER and the work of Anand and Precup (2023) are built on the intuition that learning at separate timescales helps to mitigate interference. However, the underlying logic, structural decomposition, and solution techniques are quite different. Their PT-DQN introduces an additive decomposition of the value function estimate itself, formulated as $V(s) = V_{\text{perm}}(s) + V_{\text{trans}}(s)$. In their work, both components aim to minimize the same prediction error metric but operate over different horizons to address the stability-plasticity dilemma inherent in Continual RL. The main mechanism for their coordination is the additive combination of their outputs.
>
> SCORER targets the compositional structure of Deep RL, specifically the interaction between representation ($f_\phi$) and control ($Q_\theta$), where $Q(s) = Q_\theta(f_\phi(s))$. So, rather than summing two value estimates, we model the two networks as players in a hierarchical Stackelberg game. They are intricately connected through a bi-level optimization structure where the relations between their learning rates are governed by the theoretical requirements of two-timescale stochastic approximation.
>
> Regarding the choice between methods, we view them as solving orthogonal problems and can even complement each other. Anand and Precup's approach is ideal for non-stationary task distributions where fast adaptation and retention are both key. SCORER is designed to stabilize the optimization dynamics of deep value-based learning within a single task or stationary distribution. Interestingly, SCORER is able to support the value function decomposition proposed by Anand and Precup *within* the Leader's optimization problem. Formulating this combined hierarchical optimization is a promising direction for future work, and we will incorporate this discussion into the Related Work section.
>
> **Relation to Actor-Critic and Garcin et al. (2025)**
>
> We appreciate the reference to Garcin et al. (2025), which also provides an interesting empirical investigation into the natural specialization of separate Actor and Critic networks. However, SCORER differs fundamentally by establishing a bi-level optimization setting rather than a purely architectural separation.
>
> While Garcin et al. (2025) study how separate networks tend to specialize, SCORER enforces a specific functional and hierarchical relationship via the Stackelberg interaction. The use of the Two-Timescale Stochastic Approximation creates a game where the fast-adapting Follower (representation) continuously tracks the needs of the slow-updating Leader (control). In fact, our Synchronous Coupling ablation study (Figure 6, right) studies how architectural separation without hierarchical structure performs against SCORER and a monolith baseline. Even though this separation slightly outperformed the baseline, we observed that the hierarchy further enhances performance and sample efficiency.
>
> Please note that we have revised the **Related Work (Section 2)** by adding two citations Anand & Precup (2023); Garcin et al. (2025), and clarifying that SCORER's internal game-theoretic coupling is different from additive decompositions and descriptive architectural separation.

---

> > ### Comment · Reviewer_HaiV · 2025-11-23
> > **Re: Rebuttal**
> >
> > I thank the authors for their responses: the reasoning regarding both the Anand and Precup work as well as the Garcin et al. work makes sense. I maintain a strong opinion of the work, and will engage with other reviewers during discussion with this opinion in mind.
> >
> > I will spend some additional time during the rebuttal reading the other reviews and rebuttals in more detail as well.

---

> > > ### Author Response · Authors · 2025-12-03
> > >
> > > We are happy to hear that our responses helped clarify your questions.

---

> ### Author Response · Authors · 2025-11-23
>
> **Extending SCORER to Actor-Critic and Policy Learning (OQ.1)**
>
> We anticipate that SCORER can be extended to Actor-Critic (AC) algorithms in two possible formulations. The most straightforward formulation would preserve the single-leader form, where the "Control" player would have both Actor and Critic heads. In that case, the Leader would optimize the joint AC objective on the slow timescale, while the Perception network (Follower) would minimize the variance of the Critic's Bellman error. Since the stability of value estimates is a necessary condition for the stability of policy gradients, the Follower would indirectly regularize the policy learning by conditioning the Critic's loss. Another possible route (leveraging the work of Garcin et.al (2025)) would be to instead treat Actor-Perception and Critic-Perception as two different Stackelberg games, each enforcing the specific type of stability necessary for each component.
>
> **Revision regarding Typos and Convergence Analysis**
>
> We fixed the repeated paragraph in the Introduction. We are also grateful for your positive feedback on the convergence analysis in Appendix M and have made the additional revision to state the convergence result as a formal mathematical theorem (**Theorem M.1**), listing all assumptions.

---

### Official Review · Reviewer_pMYo · 2025-10-29

**Soundness:** 2
**Presentation:** 2
**Contribution:** 2
**Rating:** 2
**Confidence:** 3

**Summary:**

The paper proposes SCORER - a method that splits deep Q-learning into two separate networks that learn at different speeds to address training instability. The representation network updates faster and minimizes Bellman error variance, while the Q-function updates slower for stability. They test on MinAtar and MiniGrid environments where some tasks show good improvements while others show more modest gains.

**Strengths:**

- The paper tackles a real issue in deep Q-learning where representation learning and value learning are tightly coupled, causing training instability.
- The learning curves show SCORER produces smoother, more stable training compared to baselines. The curves don't have as many sudden drops or erratic behavior.
- The appendix provides mathematical analysis based on two-timescale stochastic approximation. The convergence guarantees and game-theoretic formulation seem reasonable.
- Results are strong in some cases. MinAtar Breakout shows roughly 3x improvement over baseline DQN. MiniGrid Four Rooms is particularly impressive - baseline R2D2 completely fails while SCORER achieves 97% success

**Weaknesses:**

- Experiments only cover 4 simple MinAtar environments and basic gridworld tasks. No full-scale Atari with high-resolution images. This is a problem because representation learning challenges are fundamentally different at scale, and we can't tell if benefits hold with deep convolutional networks.
- Outside Breakout and Four Rooms, gains are small. Asterix is basically the same as baseline, Freeway has identical final performance (just faster), SpaceInvaders shows about 10-15% improvement. The claim of "consistently improves final performance" oversells what the data shows.
- The method uses 5:1 ratio (follower:leader learning rates) but there's no ablation testing 2:1, 10:1, or other values. This is important - we need to know how sensitive the method is and whether 5:1 is actually optimal.
- A monolithic network with higher learning rate for encoder layers (without network separation or stop-gradients) would test whether the complex game structure is necessary or if just updating representations faster works. This might not be essential since current ablations show hierarchy matters, but it would be useful to rule out simpler explanations.

**Questions:**

1. Why were full-scale Atari experiments not included, and do you have any preliminary results on high-dimensional visual domains?
2. Can you provide an ablation study on different follower-to-leader learning rate ratios (e.g., 2:1, 10:1, 20:1) beyond the 5:1 used in all experiments?
3. Have you tested a simpler baseline where a standard monolithic network uses higher learning rates for encoder layers without network separation or stop-gradients?

---

> ### Author Response · Authors · 2025-11-21
>
> We thank the reviewer for their feedback. We would like to address several points where we believe there may be some misunderstanding and report additions we included in the revised version to address some of their concerns.
>
>
>
> # W1 & Q1
>
> We have added **Section 5.2** to the revised version, where we evaluated the performance of the integration of SCORER and PQN [Gallici et al., 2025] on the **Atari-5 benchmark** in [Aitchison et al., 2023] in full Atari environments. Similar to our previous experiments, we use identical hyperparameters and, in this case, deep convolutional networks. The results in this section demonstrate that SCORER PQN matches or exceeds the PQN baseline across all scenarios, indicating that our game-theoretic coupling scales to high-dimensional visual control with deep convolutional networks.
>
>
>
> # W2
>
> We appreciate the reviewer's feedback and will adjust our assertions; however, we would like to highlight two important angles of improvement that go beyond final returns:
>
>
>
> **Sample Efficiency:** In **Appendix G** (Tables 15-18), MinAtar environments where performance is comparable, SCORER shows substantial improvements in sample efficiency (how fast agents learn to reach certain performance). In fact, these benefits are even more evident in the partially observable environments, as in MiniGrid tasks **(Section 5.3)**, SCORER's Time-to-Threshold (TTT) achieves improvements of 25% to 65% with no increase in computational cost.
>
>
>
> **Stability (New data in the revised version):** Beyond sample efficiency, **Section 5.4** demonstrates SCORER's stability advantages on challenging tasks for Q-functions. For Baird's counterexample (a classic example of the deadly triad), the Standard Linear TD diverges, where SCORER remains stable and convergent. For Stochastic Deep Sea, SCORER approaches 100% solve rates while the Bootstrapped DQN baseline reaches around 20%. We consider these to be not small improvements; they represent qualitative differences in learning reliability.
>
>
>
> **Final Performance**: Where asymptotic performance differs, SCORER offers statistical and practical advantages: In MinAtar Breakout, SCORER operates 3.4x better than baseline DQN (65.69 vs. 19.16). In SpaceInvaders SCORER operates ~16% better in final returns (148.71 vs. 127.78). Such improvements replicate across the DQN family (DDQN, DuelingDQN, DuelingDDQN) as noted in Table 1. In MiniGrid, SCORER R2D2 achieves a 97% success rate on the Four Rooms task, where baseline R2D2 achieves 0% success, and Dynamic Obstacles achieves 73% success, but SCORER R2D2 raises that to 83%. All comparisons made are with 30 independent seeds and statistical significance via Welch's t-test (p <.05). Most importantly, SCORER achieves such improvements without computational overhead.
>
>
>
> We revised how we presented our contribution to be more accurate towards "SCORER consistently either improves final performance or sample efficiency (reaching higher return in fewer timesteps)" accurately reflects the multiple dimensions of improvement demonstrated in our experiments.
>
>
>
> # W3 & Q2
>
> We appreciate this observation and, as a result, we have added **Appendix K (Figure 12)** to the revised version, including this ablation study. Here, we tested follower learning rates ranging from $2×10^{-4}$ to $1×10^{-3}$ (corresponding to ratios of 2:1 to 10:1) while fixing the leader's rate. This demonstrates that SCORER is not hypersensitive to the exact ratio; however, performance drops noticeably at the lowest ratio (2:1), confirming the necessity for sufficient timescale separation for the follower to compute an effective best response.
>
> # W4 & Q3
>
> We appreciate this suggestion. As the reviewer pointed out, our **Synchronous Coupling** ablation in **Section 5.5** (Figure 6, right plot) directly addresses the question of whether architectural separation alone (without the hierarchical Stackelberg dynamic) accounts for SCORER's improvements. The results show that this synchronous separation provides only a marginal benefit (not statistically significant), while introducing the Stackelberg hierarchy yields SCORER's benefits.
>
> Furthermore, we note that the suggested monolithic baseline (per-layer learning rates without stop-gradients) retains the direct gradient between the non-stationary Q-targets and the encoder. Since our ablation demonstrates that even **explicit** gradient separation (via stop-gradients) is insufficient without the Stackelberg interaction, it implies that simply modulating learning rates in a fully synchronized architecture would not resolve the underlying instability.
>
> [1] Matteo Gallici, Mattie Fellows, Benjamin Ellis, Bartomeu Pou, Ivan Masmitja, Jakob Nicolaus Foerster, and Mario Martin. Simplifying deep temporal difference learning. In The Thirteenth International Conference on Learning Representations, 2025

---

> ### Comment · Reviewer_pMYo · 2025-11-22
> **Response to Authors**
>
> I thank the authors for their rebuttal. However, I still have some doubts which I list down below:
>
> 1. Looking at the figure in section 5.2, I think the claim "SCORER PQN matches or exceeds the PQN baseline across all scenarios" is misplaced. I do see an improvement for Qbert and Phoenix but for others its either worse or statistically insignificant.
> 2. The authors fundamentally misunderstand my request for the simple per-layer learning rate baseline. I asked for a monolithic network with higher encoder learning rates without stop-gradients or network separation, a trivial code change. They instead tested Synchronous Coupling (which includes stop-gradients and architectural separation) and argue this "implies" the simpler baseline won't work because it retains "direct gradients between non-stationary Q-targets and the encoder." This logic is backwards: their Synchronous Coupling result could equally mean that stop-gradients are harmful when not paired with timescale separation, and that direct gradients with per-layer LRs might actually work better by preserving beneficial co-adaptation. I am not sure if I can convince myself with the current explanation.
> 3. For learning rate experiments, I am satisfied and I find the results convincing. Therefore I am willing to go from 2 to 4.
>
>
> Overall, I have decided to increase my score to 4 since the authors could prove with ablation that their method is fairly stable but I cannot recommend acceptance at this point since I find Atari-5 result claims slightly misleading and also the baseline experiment I requested was misunderstood.

---

> ### Author Response · Authors · 2025-11-24
>
> We sincerely thank the reviewer for their thorough engagement and for increasing their score. We appreciate their constructive feedback, which has strengthened our paper. We revised our paper to address both remaining concerns:
>
> # Atari-5 Results (Clarification)
> We acknowledge the reviewer's concern on clarity and precision of our original claim. We have revised Section 5.2 to state: *"Results in Figure 3 indicate that in BattleZone, DoubleDunk, and NameThisGame, SCORER PQN statistically matches baseline performance, while in Phoenix and Qbert it outperforms the baseline without adding computational overhead[...]"*.
>
> We believe this more accurately reflects the data without overgeneralizing.
>
> # Per-Layer Learning Rate Baseline (W4/Q3) - Addition
> We sincerely apologize for misunderstanding the reviewer's request. In the revised submission, **we added a set of experiments** with a **monolithic network with per-layer learning rates** (here we used encoders: 5×10⁻⁴, Q-heads: 1×10⁻⁴) to the ablation study regarding the Team Coupling Study **(Figure 6, Right)**. At the same time, we kept Synchronous Coupling (which includes architectural separation and stop-gradients). These are fundamentally different experiments, and we appreciate the reviewer's patience in clarifying this distinction. Thus, we have expanded our analysis including this new set of experiments as well as adding a new per-environment subsubsection in **Appendix K (Figure 13)**. These results strengthen our method's understanding further since we can observe that the monolithic per-layer LR baseline does improve over standard DQN, particularly in Breakout and SpaceInvaders, confirming that faster representation updates are beneficial. However, SCORER still outperforms this alternative (Breakout: SCORER 65.69 vs. Monolithic with different LR 40.3; SpaceInvaders: SCORER 148.71 vs. Monolithic with different LR 125.2; normalized IQM: SCORER ~0.98 vs. Monolithic ~0.85).

---

> ### Comment · Reviewer_pMYo · 2025-11-27
> **Update on your rebuttal**
>
> I thank the authors for their rebuttal, while the Atari-5 results are still not satisfactory, I think the work provides interesting insights that some audience might find useful, therefore I am happy to increase my score to a 6. (I am not sure what the issue is, I am not able to edit my reviews, but rest assured, I will increase your score as soon as this fix is made).

---

> > ### Author Response · Authors · 2025-12-03
> >
> > Thank you very much for your careful reading of our work and for being willing to increase your score to 6. We greatly appreciate your feedback and the constructive discussion during the rebuttal phase.

---

### Official Review · Reviewer_7cFX · 2025-10-29

**Soundness:** 2
**Presentation:** 3
**Contribution:** 2
**Rating:** 4
**Confidence:** 4

**Summary:**

The work introduces a new reinforcement learning method for online representation learning. The core idea is to divide the function approximator into two parts: one for learning a useful representation and another for learning to predict future returns. Each part is treated as an agent playing a Stackelberg game, maximizing a different objective while sharing the same Q-function predictor. This leads to a bi-level optimization problem, which they address by enforcing a faster learning rate for the encoder than for the predictor. Experiments on various tasks are presented to report the method's performance.

**Strengths:**

A. The submission focuses on an important problem of reinforcement learning, which is representation learning.

B. The reading flow is good, and the method is clearly presented.

C. The method is simple to implement as it simply uses two different learning rates for the encoder and the predictor, and updates the encoder and the predictor sequentially.

**Weaknesses:**

I. Some choices are suboptimal or unjustified.

   a. The choice of minimizing the variance of the temporal difference error is problematic. Indeed, the proposed method faces the double-sampling issue, yet it is not mentioned. This means the follower's objective function is biased. More specifically, $\mathbb{E}[\delta_j^2] \neq \mathbb{E}[\delta_j]^2$. Therefore, the presented algorithm will suffer from this issue in stochastic MDPs, where the variance of the temporal difference error exists, even for the optimal Q-function. Importantly, all experiments are made in deterministic environments. I suggest setting the follower's objective function to the temporal-difference error, which is an unbiased estimate.

   b. It is claimed that "This promotes stability in the learning process, a trait for mitigating the risk of divergence that was first formally demonstrated in foundational work by Baird (1995)" in Line 218. However, this claim is never justified by an empirical study on the Baird counter-example. I suggest evaluating the method on this counter-example.

   c. The choice of the follower and the leader is only justified by an empirical comparison. Adding a theoretical or an intuitive argument would strengthen the motivation behind this work.

II. The empirical analysis remains weak and can be improved.

   a. The method relies on multiple parallel environments. While this allows for faster training, it is an unrealistic assumption for real-world applications, and when a simulator is available, on-policy methods will be privileged. To fully assess the potential of the presented method, it would be useful to also have experiments in which only one environment is used at a time.

   b. As mentioned in I. a., it is unlikely that the presented method performs well in stochastic environments. Therefore, I suggest evaluating SCORER in stochastic environments, for example, with sticky actions.

   c. Two cofounding factors can be responsible for the performance increase. (i) The learning rate schedules are not fully disclosed. It would be more convincing to compare the proposed approach with the baselines using the same learning rate schedules (the one of the encoder and the one of the predictor). (ii) The gradients are clipped for the proposed approach but not for the baseline. Ablating this choice would help understand the benefit of SCORER.

   d. An analysis of the learning dynamics would provide more insights. For example, reporting the parameter norm and the srank would help better understand the method.

   e. While the interquantile-mean is reported as suggested by [1], the recommended way of reporting the confidence intervals is not followed, and no justification is provided.

   f. In Table 1, the maximum IQM performance is reported. This choice is uncommon and not motivated.

[1] Agarwal, Rishabh, et al. "Deep reinforcement learning at the edge of the statistical precipice." NeurIPS 2021.

### Remarks

1. The font size of every figure can be larger. Replacing the tables with visual plots would facilitate the analysis of the results.

2. Two citations can be added when speaking about the fact that learning appropriate representations helps with sample efficiency:

   [1] Dabney, Will, et al. "The value-improvement path: Towards better representations for reinforcement learning." AAAI 2021.

   [2] Vincent, Théo, et al. "Iterated Q-Network: Beyond One-Step Bellman Updates in Deep Reinforcement Learning." TMLR 2025.

3. The batch seems to be missing in equation 5.

4. In Figure 4, the left and middle Figures are swapped.

**Questions:**

N/A

---

> ### Author Response · Authors · 2025-11-21
>
> We thank the reviewer for this opportunity to clarify some misunderstandings and confusion.
>
> # W I.a
> We would first like to highlight that the loss functions for both the Leader and Follower are based on batch statistics calculated using experience replay. Unlike standard TD learning [Sutton & Barto, 2018, Sec. 6.1], which uses individual sample errors for online updates, SCORER uses Bellman errors to compute batch statistics (mean and variance) over batches sampled from the buffer.
>
> While we agree with the reviewer that the double-sampling issue causes trouble when using TD error, our proposed method computes the sample variance over a fixed batch. Because this randomly sampled batch is treated as a deterministic empirical distribution during the update, the gradient $\nabla_\phi \text{Var}_{B}[\delta]$ is exact with respect to the batch statistics. There is no sampling bias in the gradient computation for the batch itself. The cause of confusion might be the notation $\delta_j$ used in the paper, which looks like a TD error; yet, the target value Y comes from the target network, which follows the common practice in DQN and its variants.
>
> We also agree with the reviewer that the variance of the MSBE exists even for the optimal Q. However, our intuition behind such a loss design is NOT to eliminate the variance, which is impossible under stochasticity, as pointed out by the reviewer. The idea of minimizing the sample variance is to reduce the model's prediction inconsistency. In plain words, the Follower aims to make Q predictions consistent across the batch, even if the predictions are wrong, which nevertheless provides a stable and uniform signal to the Leader.
>
> We added experiments on **Stochastic Deep Sea** (Figure 5), a benchmark designed to test stochasticity. Contrary to the concern, SCORER solves this task with a near 100% success rate, while the Bootstrapped DQN baseline (Osband et al., 2016) struggles to pass 20%.
>
> # W I.b
> We appreciate this observation. We **added Section 5.4**, where, for **Baird's counterexample**, the standard off-policy approach diverges as expected, while SCORER maintains stability and converges, confirming the stabilization benefits of the Stackelberg Coupling.
>
> # W I.c
> Assigning the role of the Leader to the Control network and setting Perception as the Follower is grounded in the structural properties of the corresponding bi-level optimization. The ultimate goal of Q-learning is to identify the Q-function that minimizes MSBE. Comparing the optimization problems faced by the control and perception networks, it is a natural setting to have the control network's problem as the outer problem (also known as the upper-level problem in the literature) and the perception network's problem as the inner (lower-level) problem in the bi-level optimization connecting them. The leader-follower roles for the two networks in the Stackelberg game, thus in turn, reflect this relationship. We revised Appendix E and **Sec. 4.1** to clarify this in detail.
>
> # W II.a
> We want to highlight that our framework is agnostic to parallelization, and the use of vectorized environments is an implementation choice for computational efficiency (following standards like CleanRL and PureJaxRL, [Huang et al., 2022, Lu et al., 2022]) rather than a requirement of the method. It is worth mentioning that all our performance metrics are reported against **total environment steps** (summed across all actors), meaning the sample complexity comparison remains fair regardless of the degree of parallelism. To demonstrate efficacy in more conventional environments, we have added results on Atari environments (Section 5.2) where SCORER PQN matches or exceeds the PQN baselines without adding computation.
>
> # W II.b
> See W I.a. Our Stochastic Deep Sea results (Section 5.4) directly address this.
>
> # W II.c
> We agree that meticulous control over hyperparameters is a must-have to isolate algorithmic contributions, which is why we adhered to a strict parity protocol throughout all our experiments. As detailed in the revised **Appendix C**, we applied a two-stage tuning process, where we first performed a systematic grid search to find optimal configurations for the monolithic baselines based on conventional implementations (e.g., PureJaxRL, CleanRL, PQN). SCORER inherits this exact setup, using the identical learning rate and linear decay schedule for the Leader as the optimized baseline, meaning the only introduced variable is the Follower's rate to satisfy the timescale separation. Also, regarding the concern about gradient clipping, we clarify that this is applied consistently across all agents (baselines and SCORER). We have updated the hyperparameter tables in Appendix C to explicitly list the gradient norm constraints for the baselines and the SCORER counterparts, confirming that this is not a confounding factor.
>
> # Remarks
> Thanks for noticing that. We have added the reviewer's suggestions to the revised submission.

---

> ### Author Response · Authors · 2025-11-23
>
> # W II.d
> We appreciate the reviewer for this valuable suggestion. We agree that exploring internal learning dynamics can give us a clearer insight into the reasons for SCORER’s success, and studying these metrics is a great approach to quantify the effects of SCORER. **We have carried out the suggested analysis and included a new appendix section** (Appendix L: Learning Dynamics and Representation Analysis), which plots parameter norm and effective rank (srank).
>
> We find that SCORER and baseline converge to similar effective rank ( apprx. 119/128) by the end of training, but the two approaches differ in **when** high-rank representations are achieved. The baseline starts at a lower rank ( apprx. 107) and takes until around 40M steps to recover, while SCORER maintains near-maximal rank throughout. Parameter norms show faster growth for SCORER, with both methods stabilizing in the latter half of training.
>
> # W II.e
> We thank the reviewer for this suggestion. We agree that stratified bootstrap with percentile confidence intervals would be more appropriate than our current parametric approach for IQM and that it is also in line with Agarwal et al. (2021). We, as per Agarwal et al. (2021), updated all of the figures in the paper that report IQM to use stratified bootstrap.
>
> # W II.f
> We appreciate the reviewer for this comment. As the reviewer suggested, Table 1 now reports the IQM averaged over the final 10% of training, typical for reporting converged performance [Machado et al., 2018]. We evaluate with IQM with bootstrap confidence intervals as in Agarwal et al. (2021). All of our conclusions about SCORER remain unchanged from the original submission.
>
> [1] Agarwal, Rishabh, et al. "Deep reinforcement learning at the edge of the statistical precipice." NeurIPS 2021.
>
> [2] Machado, Marlos, et. al. "Revisiting the Arcade Learning Environment: Evaluation Protocols and Open Problems for General Agents." Journal of Artificial Intelligence Research, 2018.

---

### Author Response · Authors · 2025-12-03

Dear Reviewers and Area Chair,

We are grateful to all reviewers for their thorough, constructive, and insightful reviews. Your comments helped us to clarify important technical points, extend our experimental validations, and state our contributions more precisely. We appreciate your constructive collaboration, which enabled us to address all the concerns during the rebuttal period. In addition to clarifying several methodological points (e.g., framework and objective motivation as **7cFX's W.I.a and W.I.c**), we resolved an important **confusion regarding double-sampling bias and stochastic MDPs** (**7cFX's W.I.a**) through both theoretical clarification of our batch-based variance computation and new empirical validation, clarifying how the Follower provides consistent signals through variance minimization. We also explored fascinating questions on how SCORER relates to recent non-monolithic RL methods and how it can be extended to actor-critic approaches inspired by **HaiV's questions**. We believe these revisions and clarifications have strengthened our initial submission. Here, we provide a summary of the revisions to facilitate your review and discussion.

**Revisions (marked in blue in the paper):**

- **New Experimental Results on Full-Atari Environments** - Section 5.2 As per **reviewers pMYo's** W1/Q1, **7cFX's** W.II.a.

  - **Note:** Results in Atari-5 indicate that in BattleZone, DoubleDunk, and NameThisGame, SCORER PQN statistically matches baseline performance, while in Phoenix and Qbert, it outperforms PQN without adding computational overhead.

- **New Experimental Results and Stability Analysis with Baird's Counterexample and Stochastic Deep Sea** - Section 5.4 As per reviewer **7cFX's** W.I.a & W.I.b & W.II.b.

  - **Note:** On Stochastic Deep Sea (a benchmark designed to test Stochasticity), SCORER improves the backbone model from apprx. 20% to a near 100% success rate. On Baird's counterexample, SCORER maintains stability and converges.

- **New Ablation Studies on Appendix K: Learning Rate Sensitivity Ablation (Figure 12) & Monolithic Per-Layer Learning Rate Baseline (Figure 13)** - As per reviewers **pMYo's** W3/Q2 & W4/Q3, **b2sk's** W1.

  - **Note:** SCORER is not hypersensitive to the exact ratio; however, performance drops as learning rate ratios approach the point where they are nearly equal (e.g., 2:1), confirming the need for sufficient timescale separation for the Follower to compute an effective best response. When performing per-layer LR monolithic training, we noticed some improvement, confirming that faster representation updates are beneficial; yet, SCORER still outperforms this alternative.

- **Expanded Hyperparameter details** - as per reviewer **7cFX's** W.II.c.

  - **Note:** We perform meticulous control over all the hyperparameters used for baselines and SCORER methods, and expanded our hyperparameter details section to include the learning rate schedules we used and other remaining hyperparameters.

- **Expanded Section 4.1 & Appendix E: Role Assignment Justification** - as per reviewer **7cFX's** W I.c

  - **Note:** Assigning the role of the Leader to the Control network and setting Perception as the Follower is grounded in the structural properties of the corresponding bi-level optimization. More details in response and Section 4.1 & Appendix E: Role Assignment.

- **IQM Result Displaying (Statistical Methodology Improvements)** - as per reviewer **7cFX's** W II.e & W II.f

  - **Note:** We have updated all the corresponding figures to show stratified bootstrapped results with percentile confidence intervals, as suggested by the reviewer. Similarly, we updated Table 1 accordingly. We have also updated Appendix F to explain these metrics. All conclusions remain supported with improved statistical rigor.

- **Added new Appendix section Learning Dynamics and Representation Analysis (Appendix L)** - as per reviewer **7cFX's** W.II.d

  - **Note:** While SCORER and baseline converge to similar Srank, the two approaches differ in when high-rank representations are achieved. The baseline starts at a lower rank and takes until around 40M steps to recover, while SCORER maintains near-maximal rank throughout. Parameter norms show faster growth for SCORER, with both methods stabilizing in the latter half of training.

- **Expanded Appendix D to include analysis on Stochasticity and the Variance Objective** - complementing our insights for reviewer **7cFX's** W.I.a & W.II.b.

  - **Note:** We have added a new subsubsection analyzing SCORER's perception objective on stochastic environments.

- Revised Theorem - as per **HaiV's** suggestion.

  - **Note:** We have added a formal mathematical theorem (**Theorem M.1**) to state the convergence result, listing all assumptions.

We believe the revisions have resulted in a clearer, rigorously enhanced, and more richly empirically supported paper.

Thank you again for your valuable contributions to this work.

The Authors

---

### Meta-Review · Area_Chair_KLYw · 2025-12-29

**Summary:**

The reviewers generally found the paper’s idea of framing representation and value learning as a Stackelberg game novel, theoretically sound, and well-presented. However, several key weaknesses, including the empirical results, the justification of choices, and the completeness of analysis, remain significant.

While initial reviews were mixed (ranging from 2 to 8), the authors’ rebuttal and revisions substantially resolved most technical concerns. Reviewers ultimately agreed that:
- The conceptual novelty and theoretical rigor are compelling.
- The expanded experiments now support the paper’s core claims, even if results are modest in some tasks.
- The work makes a meaningful, well-supported contribution to the study of representation–control decoupling in deep RL.

After reading the paper, the AC can, in general, confirm the quality of this work. The consensus trend after rebuttal supports at least acceptance at the poster level (aligning with HaiV’s assessment and pMYo’s post-rebuttal upgrade).

**Reviewer Concerns:**

Concerns Effectively Addressed by the Rebuttal

- The authors provided a theoretical explanation showing that variance is computed over fixed replay batches, mitigating sampling bias. This was reinforced with new empirical results on Stochastic Deep Sea, demonstrating stability under stochastic transitions.

- The new experiments on Baird’s counterexample and Stochastic Deep Sea, empirically confirming that SCORER remains stable where conventional methods diverge.

- The added Atari‑5 evaluation with convolutional architectures, showing scalability to higher-dimensional visual tasks. Also included broader benchmarking (MinAtar, MiniGrid, and Atari).

- Clarified distinctions and connections with Anand & Precup (2023) and Garcin et al. (2025), articulating how SCORER differs structurally from additive decomposition or independent actor–critic architectures.

Concerns Still Outstanding or Only Partially Addressed:

- Although the batch-based mitigation argument is reasonable, the objective’s unbiasedness under stochastic transitions was not formally proven. The treatment remains intuitive rather than fully theoretical.

- The expanded experiments improved coverage, but large-scale or continuous-control evaluations (e.g., Mujoco, DMControl) were still absent, limiting claims of generality beyond value-based settings.

- While a monolithic per-layer LR baseline was eventually added, other straightforward alternatives (e.g., identical architecture without stop-gradients) could still yield insight and were not fully explored.

In addition, the authors also flagged an AI-generated review (b2sk), but its influence on the aggregated decision remained ambiguous.

**Reviewer Scores:**

It is worth noting that Reviewer pMYo, who initially recommended a clear rejection of the paper, later changed their assessment to a weak accept on November 28. However, the Area Chair finds the justification for this change insufficiently supported by the evidence provided. It also remains uncertain whether this shift was affected by the accidence on the OpenReview platform.

For other reviewers, b2sk is flagged as AI-generated, and the other review might not update their rating, but it does not influence the final decision of this paper.

---

### Decision · Program_Chairs · 2026-01-26

Accept (Poster)